# Ontogenetically distinct neutrophils differ in function and transcriptional profile in zebrafish

Juan P. García-López[1], Alexandre Grimaldi[2,3], Zelin Chen[4], Claudio Meneses [5,6,7,8], Karina Bravo-Tello[1], Erica Bresciani[9], Alvaro Banderas [10], Shawn M. Burgess [9] ✉, Pedro P. Hernández [11,12] ✉ & Carmen G. Feijoo [1,12] ✉

The current view of hematopoiesis considers leukocytes on a continuum with distinct developmental origins, and which exert non-overlapping functions. However, there is less known about the function and phenotype of ontogenetically distinct neutrophil populations. In this work, using a photoconvertible transgenic zebrafish line; Tg(*mpx*:Dendra2), we selectively label rostral blood island-derived and caudal hematopoietic tissue-derived neutrophils in vivo during steady state or upon injury. By comparing the migratory properties and single-cell expression profiles of both neutrophil populations at steady state we show that rostral neutrophils show higher *csf3b* expression and migration capacity than caudal neutrophils. Upon injury, both populations share a core transcriptional profile as well as subset-specific transcriptional signatures. Accordingly, both rostral and caudal neutrophils are recruited to the wound independently of their distance to the injury. While rostral neutrophils respond uniformly, caudal neutrophils respond heterogeneously. Collectively, our results reveal that co-existing neutrophils populations with ontogenically distinct origin display functional differences.

Hematopoiesis is a complex process, not only because it leads to the generation of multiple cell types with vastly different functions, but also because at specific times of development cells with the same function are generated at distinct anatomical sites and from distinct progenitor cells. The mechanisms by which different progenitor cells generate the same cell types as well as their functions during the lifetime of the organism are poorly understood. A deeper understanding of these questions will provide a better framework for the study of genetic or environmental perturbations on fetal, pediatric, and adult hematopoietic physiology.

In mammals, hematopoiesis is currently considered a continuous process, termed the "layered hematopoiesis process", in which the

[1]Fish Immunology Laboratory, Faculty of Life Science, Andres Bello University, Santiago, Chile. [2]Stem Cells & Development Unit, Institut Pasteur, 75015 Paris, France. [3]UMR CNRS 3738, Institut Pasteur, Paris, France. [4]CAS Key Laboratory of Tropical Marine Bio-Resources and Ecology, South China Sea Institute of Oceanology, Chinese Academy of Sciences, Guangzhou, China. [5]Millennium Nucleus Development of Super Adaptable Plants (MN-SAP), Santiago 8331150, Chile. [6]Millennium Institute Center for Genome Regulation (CRG), Santiago 8331150, Chile. [7]Departamento de Fruticultura y Enología, Facultad de Agronomía e Ingeniería Forestal, Pontificia Universidad Católica de Chile, Santiago 7820436, Chile. [8]Departamento de Genética Molecular y Microbiología, Facultad de Ciencias Biológicas, Pontificia Universidad Católica de Chile, Santiago 8331150, Chile. [9]Translational and Functional Genomics Branch, National Human Genome Research Institute, Bethesda, MD, USA. [10]Institut Curie, Université PSL, Sorbonne Université, CNRS UMR168, Laboratoire Physico Chimie Curie, 75005 Paris, France. [11]Institut Curie, PSL Research University, INSERM U934/CNRS UMR3215, Development and Homeostasis of Mucosal Tissues Lab, Paris, France. [12]These authors contributed equally: Pedro P. Hernández, Carmen G. Feijoo. ✉e-mail: burgess@mail.nih.gov; pedro.hernandez-cerda@curie.fr; cfeijoo@unab.cl

source of some blood lineages changes throughout the life of the animal, whereas others remain unchanged from their embryonic origin. This process takes place in three waves. In the first two waves, transient primitive and erythro-myeloid progenitors (EMPs) originate from the yolk sac. In the third wave, pluripotent hematopoietic stem cells (HSCs) originate from specialized endothelial cells within the major arteries of the developing embryo, particularly from the aorta-gonad-mesonephros (AGM) region[1–4]. It has been shown that the absence of EMPs in mice leads to embryonic lethality by E13.5 even in the presence of transplanted HCS, reinforcing the notion of a layered hematopoiesis with non-overlapping, wave-specific roles. This idea seems also to be valid for differentiated leukocytes with distinct developmental origins. For example, when yolk-derived macrophages and HSC-derived monocytes were injected to the CNS of mice lacking microglia, only yolk-derived macrophages displayed a truly microglial gene expression program[5]. Likewise, natural killer cells exhibit ontogeny-specific characteristics: EMP-derived cells exert a potent cytotoxic degranulation upon stimulation in comparison to their adult counterparts which only respond by producing inflammatory cytokines[6].

There is a lack of information about transcriptional and functional differences (and similarities) between HSC-independent and HSC-derived neutrophils. Experimental constraints to address this important gap include the spatial and temporal overlap of granulocytes emerging from HSC-independent or HSCs-dependent progenitors, and the lack of specific markers to distinguish their developmental origin.

The zebrafish provides key advantages to studying early development which have been widely exploited to study early hematopoiesis, including that of the neutrophils. In general, hematopoiesis is very similar between zebrafish and mammals[7,8]. In the first zebrafish hematopoietic wave, primitive macrophages and neutrophils develop from the rostral blood island (RBI), a hematopoietic tissue that originates from the anterior lateral-plate mesoderm located over the yolk sac[9,10]. The second wave originates in the posterior blood island (later the caudal hematopoietic tissue, equivalent to mammalian fetal liver), where HSC-independent EMPs initiate definitive hematopoiesis[11]. Finally, HSCs generated in the ventral wall of the dorsal aorta (tissue equivalent to the mammalian AGM) seed the caudal hematopoietic tissue (CHT) and give rise to all adult blood cells[11,12]. Of note, it has been shown that RBI-derived and CHT-derived neutrophils coexist during zebrafish embryonic and larval stages and that both respond to harmful stimuli[10]. However, whether neutrophils with different developmental origins have distinct molecular and functional features remains unknown.

In the present work, we perform live imaging and single-cell transcriptomics of populations enriched in RBI-derived and CHT-derived neutrophils and identify similarities and differences between these two ontogenetically distinct populations at steady state and upon tissue injury. We show that RBI-derived neutrophils have a considerably higher migration capacity than CHT-derived neutrophils at steady state, concomitant to an upregulation of *csf3b*. Upon injury, RBI-derived neutrophils respond uniformly, in contrast to CHT-derived neutrophils which show a heterogeneous response. In addition, both populations display common core transcriptional profiles, but also subset-specific transcriptional signatures. Accordingly, only CHT-derived neutrophils are recruited to the wound in a hydrogen peroxide-dependent manner. Altogether, we provide evidence that ontogenically distinct neutrophils that co-exist in the organism show key functional differences.

## Results
### Strategy to distinguish RBI-derived from CHT-derived neutrophils
We designed a strategy to be able to simultaneously distinguish between RBI and CHT derived neutrophil populations in vivo. For this,

we used the Tg(*mpx*:Dendra2) transgenic reporter line[13], in which the Dendra2 protein is expressed under the control of the myeloperoxidase (*mpx*) gene promoter. The *mpx* gene has been widely used as a neutrophil-specific marker in zebrafish[14], and Dendra2 is a green-to-red stably photoconvertible fluorescent protein derived from *Dendronephthya* sp[15]. It has been shown that the Tg(*mpx*:Dendra2) line expresses fluorescent proteins at high levels in neutrophils and at low levels in macrophages[13,16]. Therefore, to analyze only neutrophils in our imaging experiments, we used settings that excluded cells expressing low levels of Dendra2.

After leaving the rostral blood island, RBI-derived neutrophils colonize and reside in different tissues, such as the head, the trunk, and the CHT[10]. On the other hand, CHT-derived neutrophils mainly remained where they were generated[11]. We initially attempted to photoconvert Dendra2+ cells in the rostral blood island at 20hpf, however, we observed that only few of them were present in this tissue at that time (Supplementary Fig. 1B, C). Therefore, in order to photoconvert a higher number of RBI-derived neutrophils and be able to further analyze them by scRNAseq and microscopy, we determined which tissues these cells preferentially migrated to and resided in. To this end, we photoconverted all the Dendra2+ cells present in the rostral blood island on the yolk at 20hpf and determined their location 24 h later, at 44hpf (Supplementary Fig. 2A). We found that almost no neutrophils originally present in the rostral blood island over the yolk remained in this tissue 20 h later (2.63%, 0.22 cells) and that 79.32% (6.6 cells) of them were now located in the head, 10.74% (0.89 cells) in the CHT, 3.89% (0.32 cells) in the dorsum, and 3.42% (0.28 cells) in the tail (Supplementary Fig. 2B, C). Based on these results, our strategy consisted of photoconverting *mpx*:Dendra2+ cells in the head to obtain cells enriched in RBI-derived neutrophils. In these larvae, green-labelled cells should be enriched in CHT-derived neutrophils, however, since we observed 10.74% (one cell in some larvae) of RBI-derived neutrophils were present in the caudal hematopoietic tissue at 44hpf, some could potentially be of primitive origin. To avoid this scenario, we set out to have the maximum number of RBI-derived neutrophils in the head, and the minimum in the caudal hematopoietic tissue. Based on prior reports[10], we photoconverted the head at 34hpf but noted that new green neutrophils appeared in the head during the next 10 h (Supplementary Fig. 2D–F). Therefore, we determined if some of the new neutrophils were originally present in the caudal hematopoietic tissue, finding that this was never the case and confirming that they arose in the rostral blood island, (Supplementary Fig. 2G–I). Thus, to obtain the maximum number of red-labeled, RBI-derived neutrophils, we decided to perform two successive photoconversion events in the head, at 34 and 44hpf. Taken together, we established a strategy that allowed us to discriminate between a population largely enriched in RBI-derived neutrophils and one composed mainly of CHT-derived neutrophils.

### RBI-derived neutrophils have higher migratory capacity in homeostasis
We observed a constant migration of RBI-derived neutrophils from the yolk to the head and that some CHT-derived neutrophils were located outside this territory during homeostasis. To analyze the migration capacity of RBI-derived and CHT-derived neutrophils in the absence of injury, we determined what percentage of each population leaves its original location. To this end, we photoconverted the head or the CHT at 34 hpf and analyzed at 44 hpf, ten hours later, the number of RBI-derived and CHT-derived neutrophils located outside their corresponding original region (Fig. 1A). We found that 29.39% (9.4 cells) of the photoconverted RBI-derived neutrophils left the head and only 10.30% (2.47 cells) of CHT-derived neutrophils left the caudal hematopoietic tissue (Fig. 1B). Considering the difference in the migration capacity seen between the RBI-derived and CHT-derived neutrophils, we wondered whether this capacity determines which tissues each

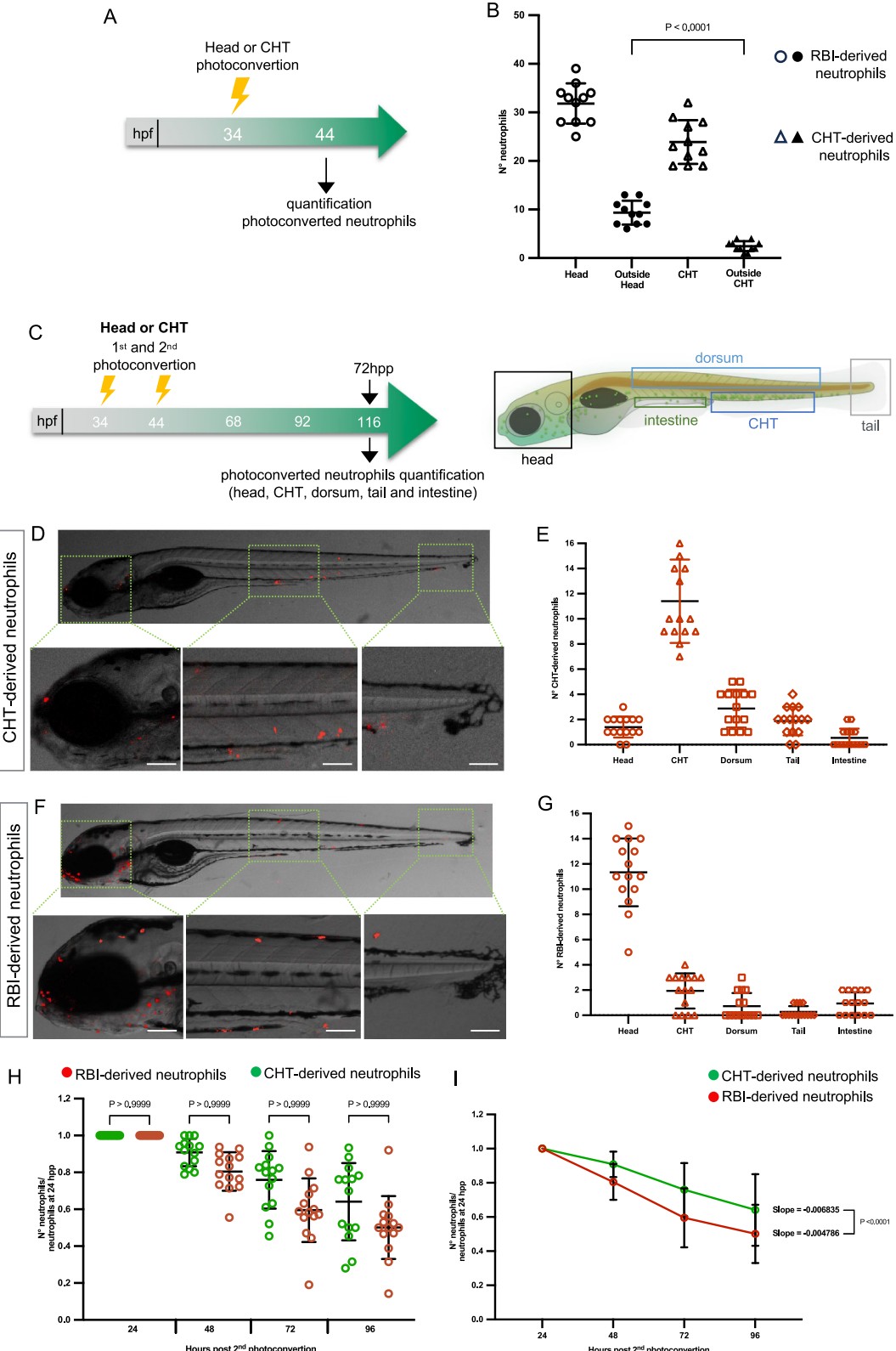

neutrophil population ultimately colonized. To allow sufficient time for neutrophils to reach regions far distant from their original position (head and posterior region of the tail), we analyzed their distribution 72 h post the second photoconversion (hpp) (Fig. 1C). We found that RBI-derived and CHT-derived neutrophils colonized the dorsum, tail and intestine. Also, a few CHT-derived neutrophils colonized the head, and a few RBI-derived neutrophils colonized the caudal hematopoietic tissue. From the 24 ± 4 red labeled CHT-derived neutrophils originally

present in the caudal hematopoietic tissue, most remained there (12 ± 4 cells), but some colonized the dorsum (3 ± 2 cells), tail (2 ± 1 cells) and head (2 ± 1 cells). Likewise, from the 32 ± 1 red labeled RBI-derived neutrophils originally present in the yolk, 11 ± 3 colonized the head, 2 ± 1 colonized the caudal hematopoietic tissue, 1 ± 1 the intestine, 2 ± 1 the dorsum and 1 ± 1 the tail (Fig. 1D–G). These data suggest that some neutrophils from both populations were lost during our analysis. Thus, to determine the period of time that both

**Fig. 1 | Migration capacity, tissue deployment and lifespan of RBI-derived and CHT-derived neutrophils during homeostasis. A** Experimental strategy design. At 34 h post fertilization (hpf) the head or the CHT was photoconverted and at 44 hpf the number of RBI-derived and CHT-derived neutrophils located in or outside the corresponding original region, was quantified. **B** Quantification of the number of RBI-derived and CHT-derived neutrophils present in and outside the head or CHT, $n = 11$ embryos for head or CHT photoconversion per experiment. **C** Experimental design. At 34 hpf and 44 hpf the first and second photoconversion were performed either in the head or CHT according to the experimental needs. **D**, **E** Representative images and quantification of CHT-derived neutrophils at 72 h post second photoconversion (hpp) present in the head, CHT, dorsum, tail, and

intestine. $n = 15$ larvae for head or CHT photoconversion per experiment. **F**, **G** Representative images and quantification of RBI-neutrophils at 72 hpp present in the head, CHT, dorsum, tail, and intestine, $n = 15$ larvae for head or CHT photoconversion. **H** Quantification of photoconverted RBI and CHT-derived neutrophils present in the whole body compared to their initial number (24 hpp), $n = 14$ larvae for head or CHT photoconversion per experiment. **I** Loss rate of RBI-derived (red line) and CHT-derived neutrophils (green line) during 72 h. $n = 14$ larvae for head or CHT photoconversion per experiment. Data are represented as mean ± SD. Statistical analyses were performed using a two-sided Mann-Whitney U test. Experiments were performed independently at least 3 times. Scale bar: 100 μm.

neutrophil populations remain in the body, we photoconverted either RBI-derived or CHT-derived neutrophils at 34 and 44hpf and monitored the number of red neutrophils present in the whole body daily during 72 h. Although some of the photoconverted neutrophils of both populations were still present in different tissues at 72hpp (Fig. 1H), the rate of loss was higher in RBI-derived (slope of −0.0047) than in CHT-derived neutrophils (slope of −0.0068) (Fig. 1I). Altogether, these data indicate that RBI-derived neutrophils have a considerably higher migration capacity in homeostasis than CHT-derived ones, but both populations colonize the same regions. Also, CHT-derived neutrophils stay for longer time period in the body compared to RBI-derived neutrophils, at least during homeostasis.

### RBI-derived and CHT-derived neutrophils display different migratory responses during inflammation

Recruitment of zebrafish neutrophils to wounded or infected areas is well documented[10,17–21]. However, whether RBI-derived and/or CHT-derived neutrophils are specifically recruited to damaged areas is unknown. To answer this question, we performed caudal fin transections 10 h after the second photoconversion (54hpf) and analyzed the recruitment of both neutrophil populations during the first three hours post damage (hpd) (Fig. 2A). As we damaged a tissue close to the caudal hematopoietic tissue, we expected CHT-derived neutrophils to migrate to the injury, but we were uncertain whether RBI-derived neutrophils would be recruited to a more distant tissue. As we anticipated, at 1 hpd, $4 ± 1.08$ CHT-derived neutrophils were attracted to the wound, increasing to $6 ± 0.88$ and $10 ± 1.55$ at 2 and 3 hpd, respectively (Fig. 2B, C). Interestingly, RBI-derived neutrophils were also recruited with $1 ± 0.71$, $2 ± 1.12$, and $3 ± 1.01$ at 1, 2 and 3 hpd, respectively, indicating a lower but consistent recruitment rate (Fig. 2B, C). To evaluate if the difference in the numbers of each neutrophil population recruited to the wound was influenced by the distance of the site of injury, we evaluated the presence of RBI- and CHT-derived neutrophils after damaging the otic vesicle, which is closer to the head. To do this, we photoconverted neutrophils at the caudal hematopoietic tissue, damaged the otic vesicle, and quantified photoconverted and not photoconverted cells in the damaged area. We observed that both populations were recruited, but in this case many more RBI-derived neutrophils reached the wound than CHT-derived ones (Supplementary Fig. 3), suggesting that proximity to the damage determines which type of neutrophils will predominantly respond. Importantly, both neutrophil populations were recruited to both injured sites regardless of the distance to the lesion.

A very early and key neutrophil chemoattractant signal released at damaged zones during acute inflammation is hydrogen peroxide[22]. Since we observed that both neutrophil populations reach the injury during the first hour after damage, we decided to evaluate if both populations were responding to this signal. To this end, we performed tail transection in the presence of diphenyliodonium (DPI), an inhibitor of NADPH oxidase that inhibits hydrogen peroxide production[23], and therefore the hydrogen peroxide dependent recruitment of neutrophils to wound. We found that CHT-derived neutrophils drastically decrease their migration to the wound during the entire period

analyzed. At 1 hpd, cells at the damaged zone declined from $3 ± 0.63$ to $1 ± 0.84$, at 2 hpd from $4 ± 1.16$ to $1 ± 0.73$ and at 3 hpd from $5 ± 1.63$ to $1 ± 0.91$ (Fig. 2D, E). In contrast, the recruitment of RBI-derived neutrophils was only affected during the first hpd, decreasing from $1 ± 0.67$ to 0 cells recruited (Fig. 2D, E), indicating that later recruitment of RBI-derived neutrophils was independent of hydrogen peroxide signaling and CHT-derived neutrophils depend on the hydrogen peroxide signal to migrate.

Next, we aimed to determine whether there are differences in the migration dynamics of RBI-derived and CHT-derived neutrophils after tail transection. It has been published that neutrophils migrate to the site of injury with marked directionality and at an average velocity of 15 μm/minute[24], however, in this study neutrophils were analyzed as a homogeneous population. To analyze the migration dynamics of both neutrophil populations separately, we performed time-lapse analysis and quantified several parameters, including pathway of migration (vasculature or extracellular matrix), directionality, cumulative and Euclidian distance, maximal velocity, and dispersion (Fig. 3). We found that CHT-derived neutrophils used both pathways of migration, as previously reported for neutrophils in general[16]. During the first hpd, CHT-derived neutrophils ($3 ± 0.75$ cells) only migrated through the extracellular matrix, but during the second and third hpd, cells transited both through the vasculature ($3 ± 0.65$ and $4 ± 0.86$ cells, respectively) and through the extracellular matrix ($4 ± 0.93$ and $2 ± 0.99$ cells, respectively). In contrast, RBI-derived neutrophils migrated to the wound only through the extracellular matrix ($1 ± 0.90$, $1 ± 0.51$, $1 ± 0.57$, respectively) (Fig. 3A, B). Only RBI-derived neutrophils present at the dorsum close to the caudal fin, and not those at the head, responded to the injury. Notably, despite some RBI-derived neutrophils circulating in the bloodstream, they never migrated towards the wound site.

Three out of the four migration parameters measured (maximal velocity, cumulative distance, directionality, and dispersion) revealed differences between RBI-derived and CHT-derived neutrophils. Only the Euclidian distance (distance between start and end points) was similar between both neutrophil populations (Fig. 3C). The maximal velocity reached by CHT-derived neutrophils was considerably higher than those of primitive ones, $135.3 ± 122.0$ μm/s and $0.87 ± 1.22$ μm/s, respectively (Fig. 3D). Of note, two clear subsets of CHT-derived neutrophils were detected. One subset including several cells that showed a fast velocity ($192.4 ± 102.7$ μm/s) and another subset including a few cells that showed a much slower velocity ($0.74 ± 0.17$ μm/s), comparable to that observed for RBI-derived neutrophils ($1.85 ± 1.3$ μm/s) (Fig. 3D). Regarding the cumulative distance, CHT-derived neutrophils traveled longer distances to arrive at the damaged zone in comparison to RBI-derived neutrophils ($352.7 ± 275.3$ μm vs. $87.53 ± 36.05$ μm) (Fig. 3E). In addition, CHT-derived neutrophils showed lower directionality in their migration to the wound compared to RBI-derived neutrophils, which was reflected in a significantly lower meandering index of the former, $0.1738 ± 0.13$ v/s $0.4044 ± 0.22$ (Fig. 3F). Dispersion represents the deviation angles of each segment of the route followed by each neutrophil with respect to the direct route to the wound (0°). CHT-derived neutrophils showed dispersion values ranging from −40° to 190° compared to RBI-derived neutrophils that

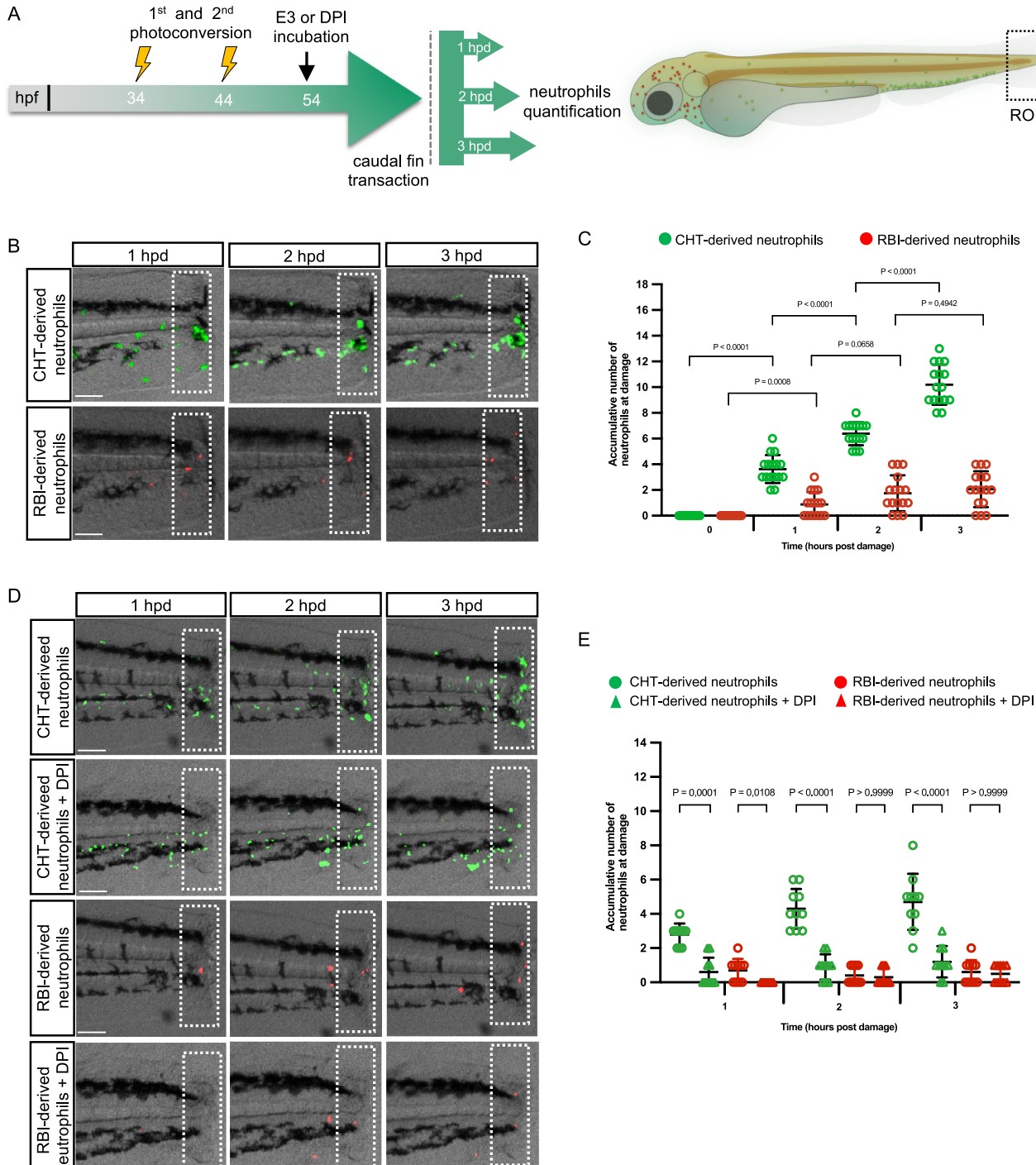

**Fig. 2 | Recruitment of RBI-derived and CHT-derived neutrophils after tail transection. A** Experimental design. At 34 h post fertilization (hpf) and 44 hpf the first and second photoconversions were performed in the head to label RBI-derived neutrophils red. At 54 hpf, caudal fin transections were made, and time lapse analysis was conducted during the 3 h post damage (hpd). **B** Representative images of RBI-derived/red and CHT-derived/green neutrophils at the wound site (dotted white line) at 1, 2, and 3 hpd. **C** Quantification of the number of RBI-derived and CHT-derived neutrophils present at the wound at 0, 1, 2, and 3 hpd. $n = 16$ larvae per experiment. **D** Representative images of RBI-derived/red and CHT-derived/green neutrophils at the wound (dotted white line) at 1, 2, and 3 hpd treated or not with diphenyliodonium (DPI). **E** Quantification of the number of RBI-derived and CHT-derived neutrophils that reach the wound at 1, 2, and 3 hpd treated or not with DPI. $n = 10$ larvae per experiment. In all the cases data are represented as mean ± SD. Statistical analysis was done with a two-sided Mann-Whitney U test. Experiments were performed independently at least 3 times. Scale bar: 100 μm.

showed values varying from −40° to only 60°. To evaluate if the dispersion of each subpopulation was homogeneous or heterogeneous, we applied the Rayleigh test to the angle distribution data. We found that CHT-derived neutrophils had a non-uniform distribution (*p*-value 0.005) while RBI-derived neutrophils showed a uniform distribution (*p*-value 0.510), indicating that the first were a more heterogeneous population than the latter. To determine whether these results were a consequence of the difference in the time at which RBI- and CHT-derived neutrophils appeared in the larva, i.e. due to differences in age, we repeated the assay of migration dynamics of RBI- and CHT-derived

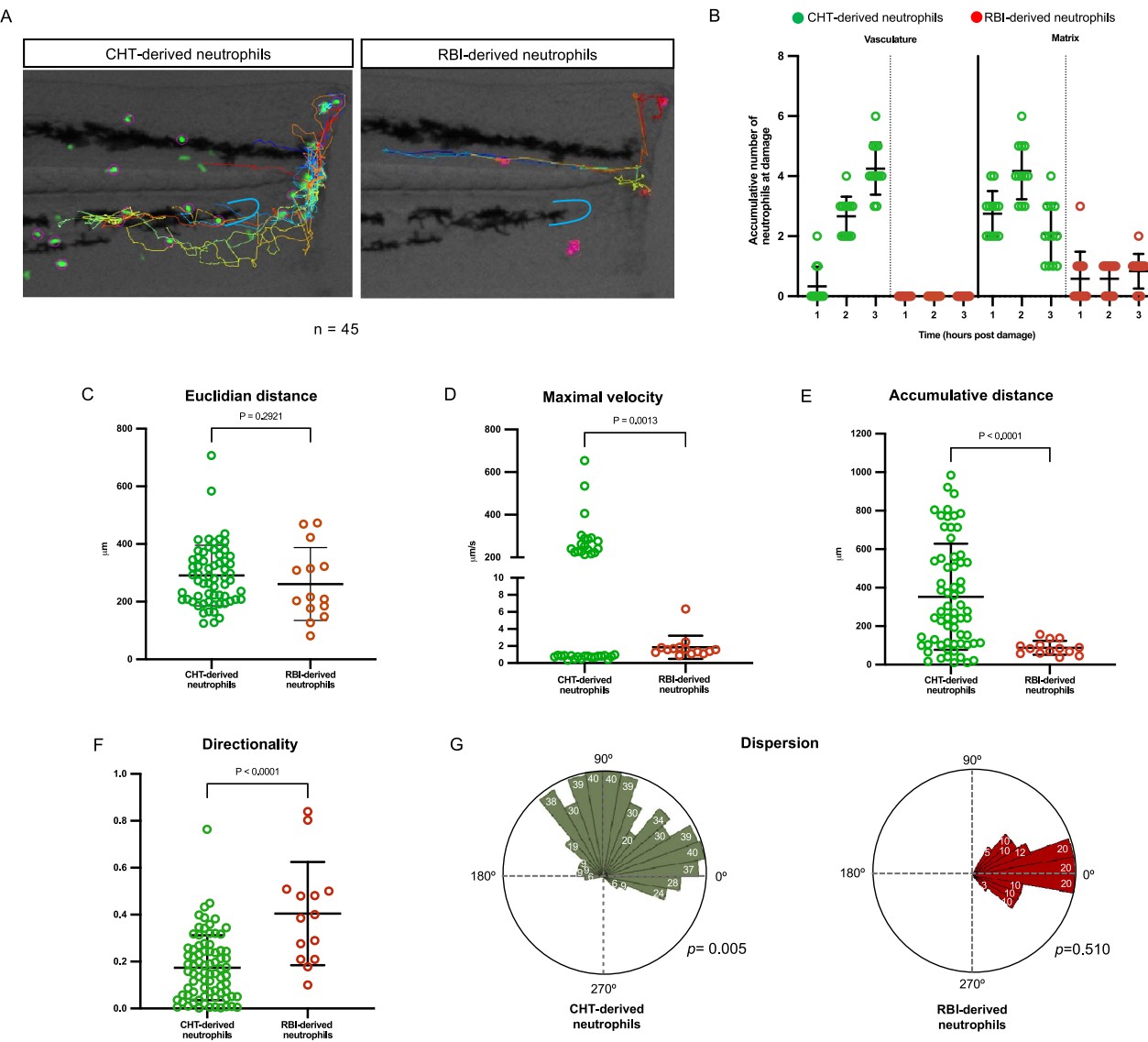

**Fig. 3 | Migration dynamic of RBI-derived and CHT-derived neutrophils during inflammation. A** Representative images of the tail at 3 h post damage (hpd) showing RBI-derived and CHT-derived neutrophil tracks to reach the wound. The blue line represents the circulatory loop. **B** Quantification of the migration route (bloodstream or extracellular matrix) utilized by RBI-derived and CHT-derived neutrophils to reach the wound at 1, 2, and 3 hpd. Analysis of the (**C**) Euclidian distance, (**D**) maximal velocity, (**E**) cumulative distance and (**F**) directionally of RBI-derived and CHT-derived neutrophils recruited to the wound. *n* = 13 larvae per experiment Statistical analyses were done with a two-sided Mann-Whitney U test. Data are represented as mean ± SD (**G**) Dispersion. Rose diagram showing the deviation angles in each segment of the route followed by neutrophils with respect to the direct route (0°) to the damage area. White numbers inside the green or red bars represent the percentage of neutrophils that follow a segment with this angle. The *p*-values were obtained after applying a Rayleigh test to angles distribution in each subpopulation. *n* = 10 larvae per experiment. Experiments were performed independently at least 3 times.

cells but 10 h earlier (Supplementary Fig. 4A). We found that both groups had the same behavior as described above in larvae 10 h older (Supplementary Fig. 4B–F). In summary, we observed that during inflammation, RBI-derived and CHT-derived neutrophils exhibited different migration behaviors. Moreover, the results obtained suggested that CHT-derived neutrophils were a much more heterogeneous population than RBI-derived neutrophils, which displayed a more uniform response.

## Single-cell transcriptional analysis of RBI-derived and CHT-derived neutrophil populations

As we observed behavioral differences between populations enriched in RBI-derived and CHT-derived neutrophils at steady state and upon inflammation, we aimed to determine if they also showed differences in their transcriptional programs in either or both conditions. To this end, we performed single-cell transcriptomic analysis of neutrophils using the droplet-based 10x Genomics platform. We photoconverted RBI-derived neutrophils in the head two times, at 34hpf and 44hpf, in 700 embryos and performed caudal fin transections at 54hpf on 350 of them. Three hours later at 57hpf, we homogenized non-cut and cut embryos respectively and separately sorted red and green-fluorescent cells. We thus defined four experimental conditions: RBI-derived normal (red cells non-cut), RBI-derived cut (red cells upon fin transection), CHT-derived normal (green cells non-cut), and CHT-derived cut (green cells upon fin transection). In total, we profiled 7028 single cells from these four conditions, detecting a median of 1211 genes per cell. Reads from the four separate conditions (flowcells) were combined and analyzed

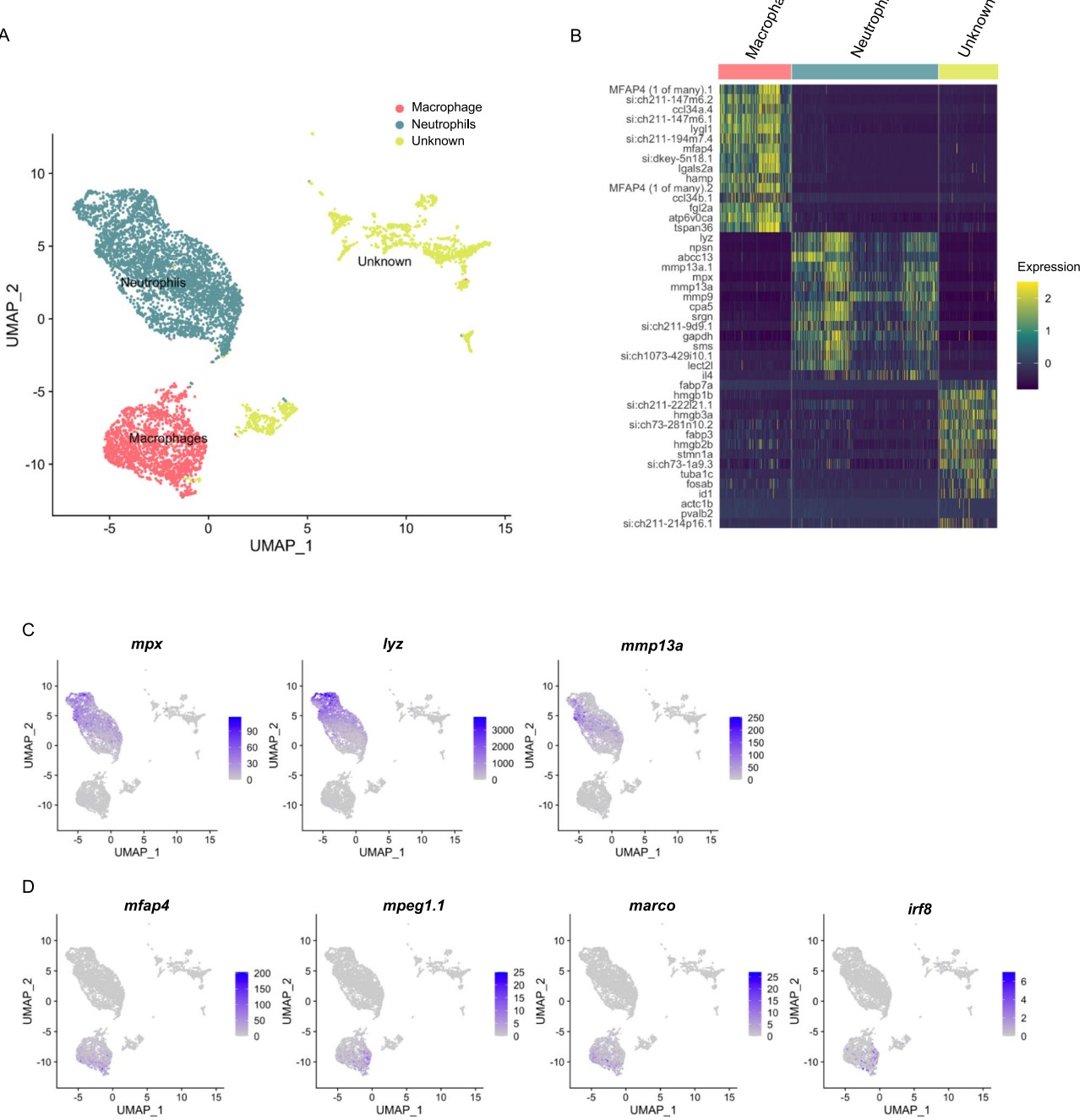

**Fig. 4 | Single-cell transcriptomic reveal neutrophil and macrophage cell populations. A** UMAP plot displaying the 3 main cell type clusters: neutrophils (3732 cells), macrophages (1830 cells), and an unknown cell type (1466 cells). **B** Heatmap of top 15 marker genes per cluster (complete list available in Supplementary Data 1, filtered on a minimum detection fraction threshold = 0.1, log fold-change threshold = 0.25 and *p*-value < 0.05 (not adjusted), negative binomial test, out of the 2000 most variable genes, only genes with positive fold change were selected). **C** Expression plots of selected markers for neutrophils: *mpx*, *liz*, *mmp13a* and *coro1a*. **D** Expression plots of selected markers for macrophages: *mfap4*, *mpeg1.1*, *marco*, *irf8*.

using Cell Ranger 2.1.1 and Seurat. We identified a major cluster composed of 3732 cells (53% of total cells), followed by a cluster of 1830 cells (26% of total cells), and a smaller cluster that altogether comprised 1466 cells (21% of the total cells) (Fig. 4A). Differential expression analysis between these 3 groups showed that the largest cluster expressed significantly higher levels (P-adj < 0.05) of 57 genes (Supplementary Data 1 "DE Lineages"), among them neutrophil markers such as *mpx, lyz, and mmp13a* (Supplementary Data 1 "DE Lineages" and Fig. 4B, C). The second largest cluster expressed significantly higher levels of 186 genes, including *mfap4.1, mfap4.2, mfap4.4, mfap4.7, mfap4.8, mfap4.10, mpeg1.1, marco* and

*irf8* (Supplementary Data 1 "DE Lineages" and Fig. 4B, C). Thus, we could identify the largest cluster as neutrophils and the second largest population as macrophages (Fig. 4B). The *mpx* gene promoter used to generate the Tg(*mpx*:Dendra2) reporter line has been shown to also label macrophages (25), explaining the capture of macrophages within profiled cells. Of note, those Dendra2⁺ cells identified as neutrophils (i.e. with high expression of *mpx*), expressed more than ten times higher *dendra2* compared to macrophages and the unknown cell group (Supplementary Fig. 5A, B). Our transcriptomic analysis also revealed markers highly expressed in neutrophils such as *ponzr6, npsn*, and the cytokines *il4* and *il34*.

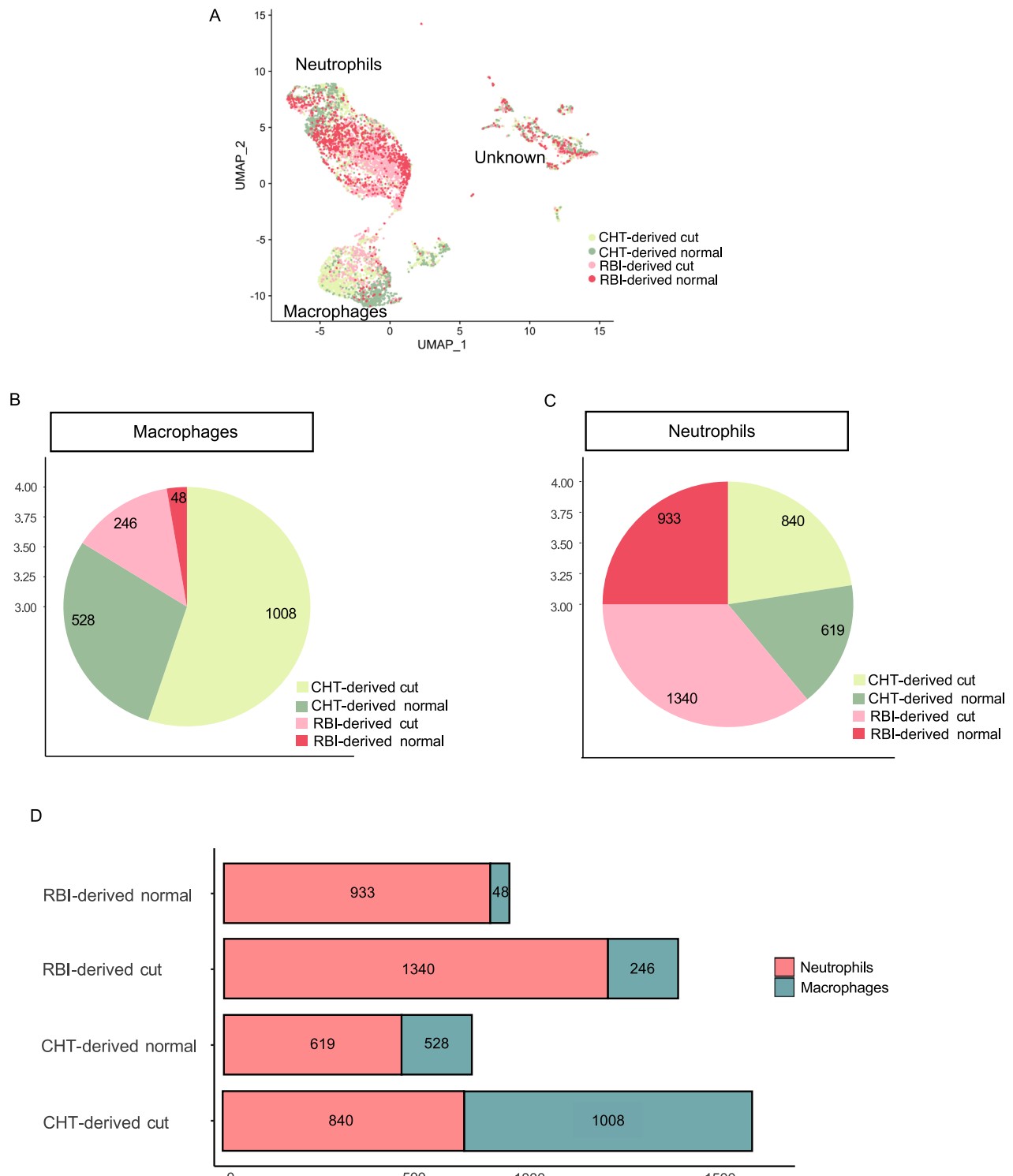

**Fig. 5 | Neutrophil and macrophage populations both contain cells from all 4 conditions. A** UMAP plot showing the distribution of cells from the four experimental conditions within the entire dataset. **B, C** Pie charts showing the number of cells of each condition present in macrophages (**B**) and neutrophils (**C**). **D** Stacked bar plot showing the number of neutrophils and macrophages present in each condition.

Likewise, we found genes highly expressed only in macrophages such as *ccl34a.4, fgl2a, il4r.1* and *il4r.2* (Supplementary Fig. 6).

We determined the contribution of cells from each condition to the neutrophil and immature macrophage populations. We found both neutrophils and macrophages in all four conditions: RBI-derived and CHT-derived cells from larvae with and without caudal fin transection (Fig. 5A). Interestingly, while neutrophils were present in similar

quantities in the four experimental conditions, macrophages were mainly present in CHT-derived cells groups (Fig. 5B, C). This implies that RBI-derived cells contributed mainly to the neutrophil population, while the CHT-derived cells contributed in a similar percentage to the neutrophil and macrophage populations (Fig. 5D). In summary, our data show that, at steady state, Dendra2⁺ cells originated in the RBI and those that migrate to the head between 34 and 44 hpf are almost

exclusively neutrophils, while Dendra2⁺ CHT-derived cells consist of neutrophils and macrophages in similar proportions. Upon inflammation, the number of macrophages increased more markedly than the number of neutrophils, with RBI-derived macrophages increasing the most.

Since we only analyzed neutrophils by live imaging (Dendra2^high cells), we focused on the neutrophil expression signature in the four conditions described above. To this end, we performed differential gene expression analyses between all the four groups but using only the data from the cluster corresponding to neutrophils, followed by gene ontology (GO) enrichment analysis of the top 300 significantly differentially expressed genes (only for the conditions where more than 300 were found differentially expressed). First, because we revealed behavioral differences between populations enriched on RBI-derived and CHT-derived neutrophils at steady state, we analyzed if there were transcriptional differences between both populations in this condition. We found that RBI-derived neutrophils expressed 5 genes at significantly higher levels than CHT-derived neutrophils at steady-state, including *csf3b*. In contrast, CHT-derived neutrophils showed higher expression of 359 genes when compared to RBI-derived neutrophils (Fig. 6A, Supplementary Data 2 "RBI-derived versus CHT-derived neutrophils steady state", Supplementary Fig. 6A).

Next, we analyzed genes differentially expressed between RBI-derived neutrophils at steady state and upon caudal fin transection. We found that 23 genes were significantly upregulated, and 121 genes downregulated upon caudal fin transection (Fig. 6B and Supplementary Data 3 "DE RBI-derived neutrophils upon caudal fin transection"). In the case of CHT-derived neutrophils, we found significant upregulation of 63 genes and downregulation of 244 genes upon caudal fin transection (Fig. 6C and Supplementary Data 4 "DE CHT-derived neutrophils upon caudal fin transection"). To compare the transcriptional changes between RBI-derived and CHT-derived neutrophils we first compared the number of overlapping upregulated and downregulated genes between RBI-derived and CHT-derived neutrophils. We observed that among the 23 genes upregulated in the RBI-derived neutrophils upon resection, a majority (20) were also upregulated in the CHT-derived neutrophils. Similarly, among the 121 genes downregulated in the RBI-derived neutrophils upon resection, a majority (94) were also downregulated in the CHT-derived neutrophils (Fig. 6D). This pronounced overlap may suggest that the basic response to injury adopted by RBI-derived neutrophils is also found in CHT-derived neutrophils, although featuring a more complex gene expression profile.

We performed network analysis of the pathways associated with upregulated and downregulated genes in RBI-derived and CHT-derived neutrophils after tail resection. Our analysis revealed that the downregulated genes of the CHT-derived neutrophils were mostly associated with translational and ribosomal processes such as "peptide biosynthetic process", "ribosome assembly", etc. Upregulated genes were mostly associated with iron sequestering and defense response including terms such as "cytokine activity", "response to lipopolysaccharide" and "neutrophil chemotaxis" consistent with immune response following injury (Fig. 6E). Upregulated genes in RBI-derived neutrophils were mostly associated with "ferroxidase activity", "intracellular sequestering of iron ion", "peptidoglycan catabolic process", "cellular iron ion homeostasis", and "chemokine activity" (Fig. 6E). Of note, *csf3b* appeared both as associated with chemotaxis upon resection and as a specific marker of the RBI-derived neutrophils when compared with CHT-derived neutrophils at steady state. Altogether, our single-cell transcriptional analysis shows that although at steady state, RBI-derived and CHT-derived neutrophils have clear differences in their transcriptional programs, they seem to share a common transcriptomic cue during injury response while retaining some diversity. After inflammation was triggered, both neutrophil populations showed a core transcriptional program, as well as subset-specific responses.

## Csf3b regulates the migration of RBI-derived neutrophils

Both live imaging and transcriptional analysis indicated an increased migration capacity for RBI-derived neutrophils compared to CHT-derived neutrophils at steady state. Among the genes showing higher expression in RBI-derived neutrophils at steady state, *csf3b* has been functionally linked to their migration capacity. This gene encodes for the Granulocyte-colony stimulating factor b (Csf3b) which regulates neutrophil egress from the hematopoietic tissue and their migration to blood vessels in mammals, and to blood vessels and tissues in zebrafish[18,25,26]. Thus, we analyzed if *csf3b* plays a role in RBI-derived and CHT-derived neutrophil migration by quantifying their egress from the corresponding hematopoietic tissues at steady state upon *csf3b* knockdown. For this purpose, we repeated the strategy shown in Fig. 1A with the addition of injecting well-established morpholinos targeting *csf3b* into the one-cell stage[18]. Then, we photoconverted either the yolk or CHT neutrophils in morphant embryos at 34 hpf and determined the number of RBI-derived and CHT-derived neutrophils outside their corresponding hematopoietic region at 44 hpf (Fig. 7A). We found that the number of RBI-derived neutrophils that egress from the yolk in morphant embryos was significantly lower than in control ones (*p*-value = 0.0038). On the other hand, morpholino injection did not affect the number of CHT-derived neutrophils staying or exiting the CHT (Fig. 7B). To explore if *csf3b* also regulates RBI-derived and CHT-derived neutrophil migration upon caudal fin transection, we performed caudal fin transections 10 h after the second photoconversion and analyzed their recruitment during the first three hpd in control and *csf3b* morphants (Fig. 7C). The number of RBI-derived and CHT-derived neutrophils that migrate to the wound was significantly lower in morphant than in control embryos at the three time points analyzed (Fig. 7D). These data indicate that the migration of RBI-derived neutrophils is regulated in a similar way during homeostasis and inflammation, suggesting a similar activation state in both conditions. In contrast, *csf3b* is only necessary for CHT-derived neutrophils recruitment during inflammation.

## Discussion

In the last decade, progress has been made in understanding how the differential developmental origin of hematopoietic cells determines their function. This has been exemplified for cell lineages such as macrophages, mast cells, and lymphocytes, however, whether developmentally distinct neutrophil populations show phenotypic and behavioral differences remains unknown. In the present work, we revealed that populations enriched in RBI-derived neutrophils and CHT-derived neutrophils display different migratory behaviors and distinct transcriptional profiles at steady state and during inflammation. However, for a proper interpretation of the presented data, it is important to clarify the nomenclature used in this work. Currently, there is no consensus on how to name the different hematopoietic progenitors during mammalian ontogeny nor in other vertebrate animal models. Nevertheless, in the case of "primitive" hematopoietic cells it seems to be simpler than for "definitive" ones. The first corresponds to a transitory hematopoietic cell population that arises from unipotent precursors immediately after gastrulation. In mammals, temporally associated unipotent precursors give rise to primitive erythrocytes, megakaryocytes, and macrophages[1,27]. In zebrafish, erythrocytes arise from unipotent progenitors, but neutrophils and macrophages arise from a common myeloid bipotent progenitor[10,28]. Thus, the neutrophil population named "RBI-derived neutrophils" in our work correspond to cells that originate from the transient HSC-independent bipotent precursors. On the other hand, definitive hematopoietic cells were traditionally considered those cells that persist in the individual

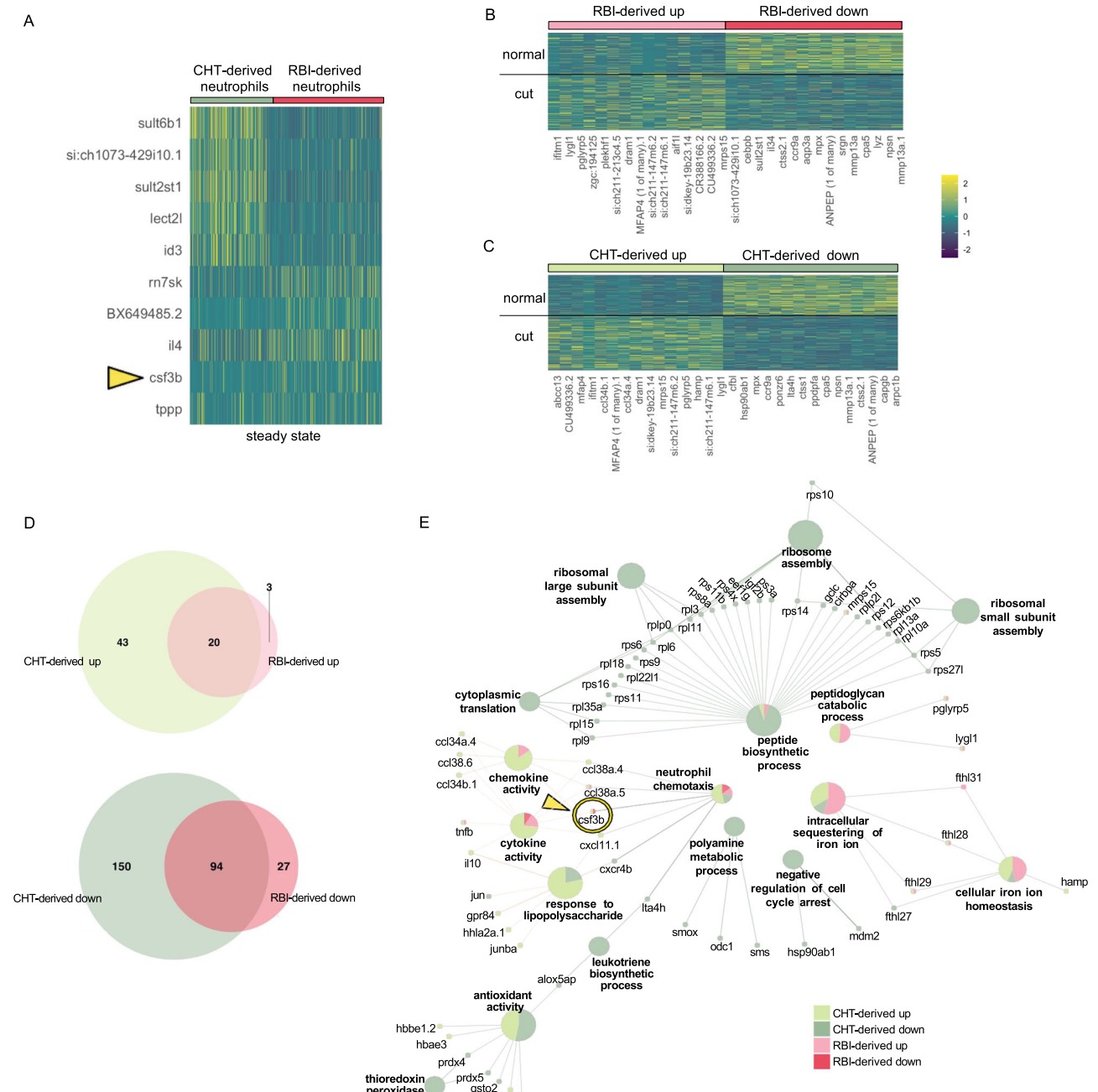

**Fig. 6 | RBI- and CHT-derived neutrophils have different responses to resection.**
**A** Heatmap of the top 5 markers in primitive and definitive neutrophils at steady state (complete list available in Supplementary Data 2, filtered on a minimum detection fraction threshold = 0.1, log fold-change threshold = 0.25 and *p*-value < 0.05 (not adjusted), negative binomial test, out of the 2000 most variable genes, only genes with positive fold change were selected). **B, C** Heatmaps of the top 15 upregulated and downregulated genes upon tail resection among the RBI- derived (**B**) and CHT-derived (**C**) neutrophil populations (complete list available in Supplementary Data 3 and 4, filtered on a minimum detection fraction threshold = 0.1, log fold-change threshold = 0.25 and *p*-value < 0.05 (not adjusted), negative binomial test, out of the 2000 most variable genes, only genes with positive fold change

were selected). **D** Venn diagram showing the number of overlapping upregulated and downregulated genes between RBI-derived and CHT-derived neutrophils. **E** Network of the pathways associated with upregulated and downregulated genes among RBI-derived and CHT-derived neutrophils upon tail resection. Each gene associated with a given pathway is displayed. Pie charts indicated the relative contribution of each type of neutrophil to that term/gene (see Methods). Note that *csf3b* appears both as associated with chemotaxis upon resection and as a specific marker of the RBI-derived neutrophils when compared with CHT-derived neutrophils. Heatmaps show genes with *p*-value < 0.05, based on a negative binomial distribution, out of the 2000 most variable genes.

until adulthood derived from HSCs[29,30]. At the present, transient HSC-independent erythromyeloid progenitors (EMP) derived from the yolk sac are also included in this category because they originate from a hemogenic endothelium[31]. In zebrafish, EMP and HSC fulfil this condition[11,12,32–34] and both populations are present in the CHT during the period of time in which the analyses were performed,

therefore, our "CHT-derived neutrophils" most likely correspond to a combination of EMP and HSC.

At steady state, our in vivo cell tracing analysis showed that more than 80% of the RBI-derived neutrophils left their hemato- poietic tissue or origin, and more than 50% of those cells colonized the head. CHT-derived neutrophils had a very different behavior

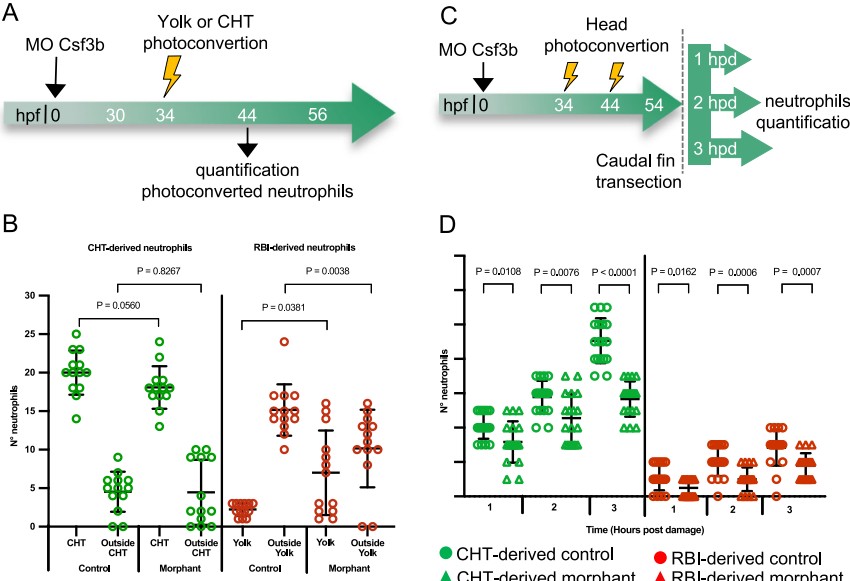

**Fig. 7 | Csf3b regulates RBI-derived neutrophil egress from hematopoietic tissue at steady state. A** Experimental strategy. At 34 hpf the yolk or CHT of control and morphant embryos was photoconverted and at 44 hpf the amount of RBI-derived and CHT-derived neutrophils was quantified. **B** Quantification of RBI-derived and CHT-derived neutrophils in and outside the corresponding hematopoietic region in control and morphant embryos. n = 13 embryos for yolk or CHT photoconvertion per experiment. **C** Experimental strategy. At 34 hpf and 44 hpf the head of control and morphant embryos was photoconverted, at 54 hpf caudal fin transection was performed and the number of RBI-derived and CHT-derived neutrophils present at the wound was quantified during the first three hours post damage. n = 18 larvae per condition per experiment. **D** Quantification of the number of RBI-derived and CHT-derived neutrophils present at the wound at 0, 1, 2 and 3 hpd in control and morphant embryos. Statistical analysis was done using a two-sided Kruskal-Wallis Test. Data are represented as mean ± SD Experiments were performed independently at least 3 times.

with only 10% of them leaving the hematopoietic tissue. This considerably higher migration ability of RBI-derived neutrophils suggests a more differentiated or activated status compared to CHT-derived ones. It is interesting to note that our transcriptomic analysis revealed that from the RBI-derived cells, almost all of them were neutrophils. Indeed, almost all the Dendra2⁺ cells expressed markers of differentiated neutrophils and just few were in early steps of differentiation expressing myeloid markers shared by neutrophils and macrophage and thus being clustered as a different population. In line with this, it has been previously reported that the population of Dendra2⁺ cells include macrophages[13]. In the case of CHT-derived cells, they give rise to neutrophils and macrophages in similar proportions, suggesting that only half of the Dendra2⁺ cells were differentiated neutrophils. This asymmetry in the number of differentiated neutrophils could be related to the dissimilar behavior observed between RBI-derived and CHT-derived cells, accounting for the higher migration capacity of RBI-derived neutrophils. The selective upregulation of *csf3b* in RBI-derived neutrophils, as well as our finding that Csf3b regulates only RBI-derived neutrophil migration and not CHT-derived neutrophils at steady state, suggest a more differentiated state for the RBI-derived cells. In mammals, CSF3 acts on neutrophils, stimulating their expansion, maturation, and release into circulation[26,35,36]. Of note, the activation of transcriptional targets in the CSF3-CSF3R signaling pathway not only depends on the binding of CSF3 to CSF3R but also on several post-translational modifications occurring in different domains of CSF3R. Two key mechanisms that negatively regulate the level, duration, and specificity of CSF3R-induced signal transduction are its internalization and lysosomal routing[37,38]. Additionally, the recruitment of suppressor of cytokine signaling 3 (SOCS3), SH2-containing inositol phosphatase (SHIP), and the tyrosine phosphatase SHP1 to CSF3R inhibits its signaling[39]. Therefore, it is possible that under steady-state conditions, CHT-neutrophils do not respond to Csf3b due to the inability of Csf3r to transmit the activation signal. In fact, it has been suggested that upon Csf3b binding to Csf3r,

the Jak2 and Pi3k signaling pathways are activated, resembling the CSF3-CSF3R signaling cascade described in mammals[40–42].

Upon fin transection, the in vivo cell tracing analysis revealed that both RBI-derived and CHT-derived neutrophils were recruited to the damaged region independently of their location and with different dynamics. CHT-derived neutrophil recruitment was controlled by hydrogen peroxide and Csf3b, in contrast to RBI-derived neutrophil migration, which was mainly dependent on Csf3b. Thus, our results indicate that Csf3b-mediated regulation of CHT-derived neutrophil migration is conserved between zebrafish and mice not only at steady state but also after injury[26,43]. On the other hand, CHT-derived neutrophils migrated to the wound throughout the extracellular matrix and the vasculature, RBI-derived neutrophils did it only across the extracellular matrix. This situation could be a consequence of differences in the anatomy of the tissues in which they are located. In the CHT, neutrophils are mainly found in the mesenchyme around the caudal vein plexus, i.e., in close proximity to the vasculature. In contrast, RBI-derived neutrophils are preferably located in the subepidermal mesenchyme of the head, more distant from blood vessels. Our transcriptomic analysis showed an overlap of GO terms associated with the upregulated genes in the two neutrophil populations with a core transcriptomic response with some level of diversity, which might be linked to the different behavior seen in our cell tracing analysis. In general, RBI-derived neutrophils behaved as a homogeneous population while the CHT-derived neutrophils were much more heterogeneous which could be related to the more complex and rich gene response of the latter population. Indeed, transcriptomic diversity may underlie this cell phenotypic heterogeneity. In a similar scenario in mammals during acute inflammation, heterogeneity has also been observed in neutrophils in circulation: mature CD16^bright CD62L^bright neutrophils with a segmented nucleus, hypersegmented CD62L^low neutrophils and CD16^low neutrophils[44]. Interestingly, these morphologically distinct populations also exerted different functions. CD16^low neutrophils exhibit a high capacity for phagolysosomal acidification and bacterial containment, and CD62L^low neutrophils display poor bacterial killing capacity, but

induced suppression of T cell proliferation[44,45]. Of note, these observations were obtained using only HSC derived neutrophils. An important point to consider in the case of the CHT-derived neutrophils is that at the time we performed our experiments this population was composed of neutrophils originated from EMP and HSC. In contrast, RBI-derived neutrophils arose from a single embryonic precursor. Therefore, it is possible that EMP-derived neutrophils and HSC-derived neutrophils could have different features.

Finally, our transcriptomic analyses revealed several genes highly expressed in neutrophils, placenta-specific 8 protein related 6 (*ponzr6*), nephrosin (*npsn*), *il4* and *il34*, which have not been previously identified as markers of granulocytes. The mammalian ortholog of *ponzr6*, *PLAC8*, is involved in various cellular processes, such as the regulation of immunity, cell differentiation, apoptosis, and tumor progression[46,47]. Interestingly, in colon cancer, PLAC8-overexpressing cells exhibited elevated cell motility and cancer invasion[48]. In zebrafish Npsn is a component of neutrophil granzymes that promotes host defense against bacterial infection[49]. Also, zebrafish Il34 regulates granulocyte differentiation[50].

Although the existence of heterogeneity among neutrophils is now recognized, whether the phenotypic variation can be explained by a different maturation level and activation status within a single cell type or if they are true different cell types is still under debate. More importantly, the biological relevance of this diversity remains to be elucidated. Thus, understanding the differences and similarities of RBI-derived and CHT-derived neutrophils will help to reveal to what extent their identities are dictated by the environment or rely on their intrinsic qualities, information that is central to exploit their use for biomedical research.

## Methods
### Fish and transgenic line
Zebrafish were maintained and bred in our facility according to standard protocols[51]. The Tg(*mpx*:Dendra2)[13] was used to perform the experiments in this study. All embryos were collected by natural spawning and maintained at 28 °C in E3 medium (5 mM NaCl, 0.17 mM KCl, 0.33 mM CaCl2, 0.33 mM MgSO₄, with methylene blue, equilibrated to pH 7.0) in petri dishes. Embryonic and larval ages are expressed in hours post fertilization (hpf) or days post fertilization (dpf). All experiments were executed with approval from the Ethics Committee of the Universidad Andres Bello.

### Photo-labelling of neutrophils
The rostral blood island, the head or the caudal hematopoietic tissue of embryos were photoconverted at different timepoints (Supplementary Fig. 1) using the protocol described previously in ref. 13 with minor modifications. Briefly, embryos were anesthetized with 0.017% tricaine and located in drops of E3 medium on a slide, then the whole head region was photoconverted with a 405 nm laser for 30 s at 50% power with an epifluorescence microscope (Olympus BX61).

### Caudal fin transection
At 54 hpf, caudal fin transection was performed to previously photoconverted embryos using the severe damage model as previously described in ref. 17. The number of neutrophils recruited at the damaged area were quantified at 1, 2 and 3 h post damage (hpd) and the migration tracked for 3 h, until 57hpf.

### Otic vesicle damage
At 48 hpf, the previously photoconverted larvae were anesthetized with tricaine and embedded in 4% methylcellulose. Three punctures were performed with a microinjection needle on the left otic vesicle and the number of neutrophils recruited to the damaged area was quantified at 0.5-, 1-, 1.5- and 2 h post damage (hpd). Also, migration of RBI-derived and CHT-derived neutrophils was tracked during the 2 h analyzed.

### DPI assay
Experiments with DPI (diphenyliodonium) (43088, Sigma-Aldrich) were performed as previously described in ref. 16. Briefly, embryos were pre-incubated for 1 h with DPI 100 µM in 1% DMSO in E3 before caudal fin transection was made.

### Cell migration tracing analysis
Cell tracing was performed with the TrackMate 3.8.0 plugin from Fiji software using an Operator by Differences of Gaussian (DoG) measurement to obtain coordinates of paths. Quantitative neutrophils behavior measurements (maximal velocity, directionally, Euclidian, accumulative distance, and spatiality) were extracted using the Chemotaxis and Migration Tool 4.3.2 software.

### Imaging and Time-lapse
Embryos and larvae were mounted in 1% low melting agarose or 4% methylcellulose. Images were obtained with a Leica M205FA stereomicroscope coupled to a Leica DFC7000T camera. For Time-lapse analysis, images were acquired every 30 s for 2 or 3 h. Images were processed with Gimp 2.10 and Fiji.

Fluorescence Activated Cell Sorting (FACS) (Supplementary Fig. 7) and Single-cell library construction and sequencing 350 embryos per condition were dissociated as described previously in ref. 52. The single cell suspensions were sorted on a FACS AriaIII (Becton Dickinson, Franklin Lakes, NJ) using FACS DIVA software. Sorted cells were first gated using the Forward Scatter Area versus Side Scatter Area dot plot. Two additional dot plots were used to remove cell clumps by using a singlet gate on Forward Scatter Area versus Forward Scatter Width followed by a second singlet gate on Side Scatter Area versus Side Scatter Width. Cells falling within these three gated regions were then separated between green⁺ (488 nm wavelength) and red⁺ photoconverted (543 nm wavelength). A total of 7000 red⁺ and 16,000 green⁺ viable cells were collected in a small volume of DMEM-10%FBS (25–30 ul) in a PCR-tube and the final volume was adjusted according to the 10X Chromium protocol.

Single-cell RNA-seq libraries were prepared using the Chromium Single Cell 3′ Library & Gel Bead Kit v2 (PN 120737, 10x Genomics). 7000 red⁺ and 16,000 green⁺ cells were loaded into the Chromium Controller instrument (10x Genomics) to generate Gel beads in EMulsion (GEMs). GEM-RT was performed in a C1000 Touch Thermal cycler with the settings: 53 °C for 55 min, 85 °C for 5 min, held at 4 °C. After reverse transcription, GEMs were broken, and the single-strand cDNA was cleaned up using DynaBeads MyOne Silane Beads (37002D, Thermo Fisher Scientific) and the SPRIselect Reagent Kit (B23318, Beckman Coulter). cDNA was amplified using the C1000 Touch Thermal cycler with the setting: 98 °C for 3 min, cycled 12 ×: 98 °C for 15 s, 67 °C for 20 s, and 72 °C for 1 min, 72 °C for 1 min, held at 4 °C. Amplified cDNA product was cleaned up with the SPRIselect Reagent Kit (B23318, Beckman Coulter) and quantified obtaining between 39 and 83.5 ng per library. Libraries were constructed using the reagents in the GemCode Single-Cell 3′ Library Kit (PN 120737, 10x Genomics) at National Institutes of Health Intramural Sequencing Center (NISC), following these steps: 1) end repair and A-tailing, 2) adapter ligation, 3) postligation cleanup with SPRIselect, 4) sample index PCR and cleanup. The barcode sequencing libraries were quantified by quantitative PCR. Libraries were then sequenced over two lanes of Illumina HiSeq 4000 with 150 bp paired-end reads.

### Single-cell RNA-seq analysis
scRNAseq datasets were preprocessed using Seurat in R (https://satijalab.org/seurat/)[53]. Cells with more than 10% of mitochondrial gene fraction, less than 300 detected genes, more than 7500 genes or less than 500 UMIs were discarded. Dimension reduction and UMAP generation were performed following Seurat workflow. Doublets were

inferred using DoubletFinder v3[54] and a nFeatures/nUMIs plot and removed before further processing. Cells expressed a median of 4655 nUMIs, and a median of 1211 genes. Cell cycle was regressed using a "CC.Difference" scoring (S.Score-G2M.Score) following Seurat dedicated vignette. Annotation of clusters and subsetting were also performed in Seurat based on the expression patterns of markers in Fig. 4C. Differentially expressed genes were determined using the "FindAllMarkers" function in Seurat with the "test.use" parameter as "negbinom". All other parameters were kept as default. In order to be consistent with the genes displayed using the "DoHeatmap" function of Seurat, we filtered the identified markers to only keep those present in the "scale.data" slot of the Seurat object. We provide those lists of markers in additional tables (Supplementary Data). Gene set enrichment analyses were performed with those markers using Cluego[55], a Cytoscape plugin. The databases used were 'GOMolecular Pathway', 'GO BiologicalProcess' and 'GO ImmuneSystemProcess' with GO Tree interval set a Min = 6 and Max = 12, using GO term fusion, and only displaying pathways with pV < = 0.05. An enrichment/depletion two-sided hypergeometric test was performed, and $p$-values were corrected using the Bonferroni step down method.

The gene expression analyses for *dendra2*, *mpx* and *csf3b* were performed by comparing the number of normalized counts between the different conditions using one-way ANOVA and Benjamini-Hochberg post-test ($q$-value < 0.05).

### Csf3 knockdown assay

Csf3 morpholino (Gene tools) was used as previously described in ref. 18. Briefly, 5 ng were injected to each embryo at 1-cell stage and at 48hpf, a decrease in the mRNA expression levels was tested by RT-PCR using the following primers flanking exon 3 and intron 3: Forward 5´-GTGAGTTCCAGATCCCGACG -´3, Reverse 5´- GTGATGAAGCTCCAG ACCG-´3.

### Statistical analysis

All experiments with the exception of the single-cell transcriptomic analyses were performed in triplicate with at least 15 individuals per condition if not specified otherwise in the figure legend. Statistical analysis was performed using Prism 6.0 (GraphPad Software). After a Shapiro Wilk normality test, the data was treated as not parametric and Fisher´s, Kruskall Wallis and Mann Whitney U tests were performed. Outliers were removed using the ROUT method with Q = 1%. Statistical significance was determined with a $p$-value of 0.05.

### Reporting summary

Further information on research design is available in the Nature Portfolio Reporting Summary linked to this article.

## Data availability

The scRNAseq data generated in this study have been deposited in the GEO database with primary accession number GSE239880. All other data are available in the article and its Supplementary files or from the corresponding author upon request. Source data are provided with this paper.

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

## Acknowledgements

This research was supported in part by the Intramural Research Program of the National Human Genome Research Institute (ZIAHG200386-06) to SMB; Fondecyt grant 1210903, DI-02-22/NUC, ECOS-ANID C22S01-220029 and Alexander von Humboldt Foundation fellowship to CGF; Fondecyt grant 1200804 and ANID-Millennium Science Initiative Program, Millennium Nucleus for Development of Super Adaptable Plants to CM; Labex DEEP (ANR-11- LBX-0044, ANR-10-IDEX- 0001-02 PSL), FRM (AJE201905008718), Ville de Paris (2020 DAE 78) and the ATIP-Avenir starting grant to PPH.

## Author contributions

J.P.G.L., S.M.B., P.P.H., and C.G.F. contributed to the design of the experiments, J.P.G.L., A.G., Z.C., K.B.T., A.B., C.M., and E.B. performed the experiments. All authors interpreted and analyzed the data. P.P.H. and C.G.F. co-wrote the manuscript. All authors read and edited the manuscript and approved the submitted version.

## Competing interests

The authors declare no competing interests.
