## [Peer Review File · Nature Communications]

Ontogenetically distinct neutrophils differ in transcriptional profile and behavior in zebrafishEditorial Note: Parts of this Peer Review File have been redacted as indicated to maintain the confidentiality of unpublished data.

REVIEWER COMMENTS

Reviewer #1 (Remarks to the Author):

During ontogeny, multiple waves of hematopoiesis (primitive wave, EMP wave, and definitive wave) successively contribute to neutrophil generation. Those ontogenetically distinct neutrophils co-exist during certain developmental period. Yet whether those neutrophils of distinct origins manifest different cellular behaviour and function remain unclear. In this manuscript, the author made use of photoconvertible Tg(mpx:dendra2) zebrafish to label the RBI-derived (primitive wave) and the CHT-derived (EMP and definitive wave) neutrophils and monitored the behaviours and transcriptomes of mpx⁺ neutrophils under homeostasis and wound (fin-cut) conditions. Although the authors showed several differences of RBI- versus CHT- derived neutrophils, e.g., migration parameters and csf3b dependency, the findings (cellular motion and response to wound) lack conceptual advance and the data presented are insufficient to conclude that the RBI- and CHT-derived neutrophils are indeed differences inters of their behaviours and functions.

Major concerns:

1. Because the RBI-derived neutrophils arise earlier than CHT-derived ones, the authors should optimize their experiment settings to exclude the possibility that the cellular difference observed in their assays is not due to the difference in terms of their maturation.
2. The conclusion “CHT-derived neutrophils have a longer lifespan compared to RBI-derived neutrophils, at least during homeostasis (Lines 184-185) is questionable. The loss of photoconverted dendra2⁺ (red) positive cells could be due to either cell death or excessive proliferation (the red fluorescence protein becomes diluted). Further in-depth characterization is needed to determine their lifespan.
3. The authors compared RBI-derived neutrophils with CHT-derived neutrophils, and they found only 5 genes upregulated in the RBI-derived cells in steady state. The csf3b gene is selected but with p-adjust value = 1 (Table 2) and the heatmap in Fig. 6A cannot clearly tell

the difference of the *csf3b* expression in the RBI- and CHT-derived neutrophils. Hence, the conclusion that “RBI-derived neutrophils show higher *csf3b* expression” needs to be carefully reinvestigated.

4. Using *csf3b* knockdown assay, the authors showed that the egression of the RBI- but not CHT-derived neutrophils is affected. However, these data are too preliminary. Whether the egression impairment is due to the developmental defect (cell autonomous vs non-cell autonomous as well) remains unclear. The authors should examine the expression of the *csf3rb* receptors in these two ontogenically different neutrophils and how RBI- and CHT-derived neutrophils respond differently to *csf3b* signaling. These questions need to be addressed for the improvement of the manuscript quality.

Reviewer #2 (Remarks to the Author):

What are the noteworthy results?

This is a very nice study, using a powerful new model to perform a detailed characterisation of phenotype and distribution of neutrophils from different developmental organs. These are characterised in detail using RNAseq.

Functional profiling is followed up with single cell RNAseq analysis of CHT and RBI sorted neutrophil populations, revealing a core programme and differentially expressed genes, some of the phenotypic features depend differentially on CSF3R function, tested by knockdown.

Will the work be of significance to the field and related fields? How does it compare to the established literature? If the work is not original, please provide relevant references.

The manuscript will be of exceptional interest to zebrafish neutrophil biologists and the ultimate conclusions of interest to neutrophil biologists more generally - if some of the weaknesses can be addressed. See below. Work of this nature has not been previously performed to my knowledge. similar experiments measuring neutrophil lifespan have been performed previously and are not cited. Dixon et al ISRN Hematology 2012:915868

Does the work support the conclusions and claims, or is additional evidence needed?

In general the experiments are well performed and the conclusions drawn support the claims made. There is one series of exceptions to this, which are discussed below.

Are there any flaws in the data analysis, interpretation and conclusions? - Do these prohibit publication or require revision?

The promoter used is a relatively short promoter and might be expected not to fully recapitulate the endogenous expression - in fact the number of macrophages identified using this approach, which do not express endogenous mpx would support this. This has consequences for much of the downstream analysis. The authors could investigate this by comparing mpx transcripts and dendra transcripts in the sequencing data and I would strongly suggest they show dendra data as a panel on Figure 4. Significant discrepancy here might undermine the generalisability of the conclusions drawn. As well as a discussion of this potential weakness in this study, the authors should filter out non neutrophils before performing the analysis in figure 6. If these data are already filtered, this should be made clear in the main text and the figure legend, as currently it would appear that macrophage genes are included in the analysis. Gene ontology should also be only performed on this filtered dataset.

Is the methodology sound? Does the work meet the expected standards in your field?

Other than the issue raised above, the work is performed using widely accepted methodologies.

Is there enough detail provided in the methods for the work to be reproduced?

Yes.

Minor issues

Line 246 - CHT neutrophils are a mixture of CHT and RBI neutrophils, so maybe this explains why there are 2 different populations? Also relevant to line 447. CHT cells more heterogenous in behaviour. Varying degrees of development as well as containing 2 populations - RBI cells migrated into CHT and native CHT cells.

Line 440. RBNI neutrophils only migrate through tissue, while CHT migrate through vessels. Obvious anatomical explanations for this which are not discussed.

Line 457 “a couple of genes” . Suggest: “several”

Line 457 - the new granulocyte genes mentioned would be stronger if cross-referenced to one of the many human RNAseq datasets available.

The colours in Figure 5 are difficult to discern and could be made clearer.

Line 295 - give rise to...is perhaps misleading. The RBI cells are exclusively neutrophils, while the CHT cells either are or give rise to.

Line 297 more pronouncedly is grammatically incorrect.

|Figure 7 - a Mann-Whitney U test would need to be corrected for multiple comparisons.

Reviewer #3 (Remarks to the Author):

The manuscript « Ontogenetically distinct neutrophils differ in behavior and transcriptional profiles in zebrafish » by Juan P. García-López, Alexandre Grimaldi, Zelin Chen, Karina Bravo-Tello¹, Erica Bresciani, Alvaro Banderas, Shawn M. Burgess, Pedro P. Hernandez and Carmen G. Feijóo addresses the interesting problem of the specificities of the neutrophils from the primitive versus the definitive wave. This question is clearly of importance for the field, and more broadly in the field of early innate immunity. They chose to address this important question using the zebrafish model, and the results presented in this manuscript suggest that these two neutrophil populations may have different flavors.

I have two categories of concerns:

Main concerns to be addressed:

1) Macrophages versus neutrophils: The “transcriptomic” part of the manuscript reveals that it is probable that the so called CHT-derived neutrophils studied in the first three sections of the results are in fact both macrophages and neutrophils (roughly 50%/50%) severely compromising the conclusions. For the study of their CHT-derived neutrophils as reported in these first three sections of the results, the authors shall therefore implement

new strategies to discriminate macrophages versus neutrophils.

2) Age of the different populations of neutrophils:

The first neutrophils of the primitive wave are present in the RBI at 20 hpf, and may be detected as soon as 16 hpf while those of the second wave are not detected before 30 hpf. When the authors do photoconvert neutrophils from both waves at 34 + 44 hpf, and analyze them at the same time point, the respective RBI and CHT derived neutrophils do not have the same “age”; the RBI derived ones are “older” ones. The authors should consider their age differences and set up experimental frames to integrate this difference to measure their respective migration capacity, lifespan, H2O2 dependent migration, migration dynamics and parameters, and transcriptomic profiles.

3) Origin versus homing:

The RBI is the place where primitive neutrophils do appear, and this is fundamentally different from the CHT that is a homing tissue where HSC from the AGM do differentiate. It is therefore not relevant to compare their capacity to “retain neutrophils”.

4) Single cell transcriptomics:

While for the first three sections of the results, the authors have work with both RBI- and CHT-photoconverted cells, in the “transcriptomic” section, for obvious financial and practical reasons, the authors have decided that photoconverted cells are from RBI origin while non photoconverted ones are from CHT origin. If the corresponding results are true, then half of the CHT-derived cells analyzed in the previous parts of the article are in fact macrophages while only a small fraction of the RBI-derived cells are macrophages. This could interfere severely with the conclusions draw from data presented in figures 1 to 3. It is therefore necessary to perform this transcriptomic analysis with CHT-photoconverted neutrophils not only to be sure that the population labelled as “CHT-derived” is of true CHT origin, but also to know the proportion of macrophages in the CHT-derived populations analyzed in the previous sections of the manuscript (figures 1 to 3)

Minor concerns to be addressed:

5) Figure to highlight what RBI is! It would be nice to have a first figure panel for this

6) Lines 130-131: To my opinion, reference 23 is not relevant for the assertion “On the other hand, CHT-derived neutrophils mainly remain in the caudal hematopoietic tissue (23)”. The work presented in the manuscript shows that this assertion is correct for the first 48 hours of development.

7) Line 147: “...new green neutrophils appeared in this area...” it is not clear what “this area” is. Is it the head or the CHT? Please clarify in the text.

8) Lines 146-150: This part is dedicated to setting up the photoconversion frame so that a maximum of RBI derived neutrophils would be labelled. If I understood well, the authors show that green neutrophils that appear in the head between 34 and 44 hpf do come from the RBI (primitive wave). Therefore, green neutrophils present in the head at 44hpf in panel S1E, are not “CHT-derived neutrophils”. Panel S1E shall therefore be changed and “CHT-derived neutrophils” replaced by “green neutrophils”.

9) Figure Sup1:

The size of the fonts is too small, it is difficult to read, and when enlarged, they pixelize.

The ordinate of panel B is absolutely not adapted for two reasons: 1) For 20 hpf, as the count of neutrophils is 0, it is not possible to use a % scale. 2) panels E and H are with absolute numbers, it is necessary to homogenize so that the reader could compare. Please use absolute numbers for panels B, E & H.

The color code for panel E does not allow to recognize RBI versus CHT derived cells. The dots for RBI derived neutrophils appear black, it would sound obvious that they would be red.

Panel H: The ordinate is labelled “number of neutrophils” while it should be labelled “number of red neutrophils”... and it would be much easier to understand with red dots.

10) Lines 160-162: For this part of their work, the authors consider that green macrophages are CHT-derived while red ones are RBI derived. This is not acceptable because the authors have explained in the previous paragraph that green cells in the head at 44hpf were in fact of RBI origin (lines 147-150).

Figure 1B shall therefore be changed.

11) Line 161 “photoconverted the head or the at 34 hpf and analyzed at 44 hpf”, it seems that the authors forgot at least a word after “or the”.

12) Panel 1C is supposed to be a graphical abstract of the time frame of the experiment, but the scale is hpf while data 1D to 1I are at 72 hours post second photoconversion (hpp). Please indicate 72 hpp in panel 1C.

13) Panel 1E, “neutrophils” should be replaced by “CHT-derived neutrophils”

14) Panel 1G, “neutrophils” should be replaced by “RBI-derived neutrophils”

15) Figure 2 panel A (same in S2A): The graphical abstract of the experimental strategy is misleading. The green arrows for 1hpd, 2 hpd and 3 hpd should be of different lengths.

16) Line 214 “... diphenyliodonium (DPI), an inhibitor of NADPH oxidase that inhibits hydrogen peroxide signaling”; DPI does not inhibit signaling, it inhibits production of H₂O₂.

17) Line 215 “... and therefore the recruitment of neutrophils to wound”; No! the sentence should be “... and therefore the H₂O₂ dependent recruitment of neutrophils to wound”

18) Lines 263-339: The exact time frame of the experiment is lacking. The time of transection is lacking, and the number of hours between damage and homogenization of embryos for cell sorting is also lacking.

19) Lines 291-292 “Interestingly, while neutrophils contained a similar percentage of cells from all four conditions, macrophages mainly comprised CHT-derived cells (Figure 5B)”. This sentence is not understandable! It should be rephrased.

20) Line 295 “..., mpx:Dendra2+ RBI-derived cells give rise almost exclusively to neutrophils ...” No!

The statement should be “..., at 34 and 44 hpf mpx:Dendra2+ cells in the RBI are give rise almost exclusively neutrophils ...”

21) Line 296 the statement “...mpx:Dendra2+ CHT-derived cells give rise to neutrophils and macrophages in similar proportions.” is wrong! This is due to the fact that the authors consider all green cells as CHT derived.

22) Line 297 “Upon inflammation, macrophages increase more pronouncedly than neutrophils...” What exactly the authors mean by “increase” is it cell size or volume or is it numbers? ... does this mean that macrophages have a longer lifespan or that they replicate while neutrophils don’t and even die?

23) Line 280 “The second largest cluster expressed significantly higher levels of 186 genes, including mfap4...” while in tables 2 and 3 the authors show that the different populations of neutrophils express different levels of mfap4. As the authors highlight, there are many different mfap4 transcripts, it is therefore necessary that the authors refer to reference zebrafish mfap4 transcripts so that the reader would know exactly which transcript is being considered. Furthermore, as the mfap4 expression is considered as an excellent marker for macrophages, it is surprising that abundance of mfap4 transcripts could vary between different neutrophil populations. It would be very helpful for the reader to have an idea of the relative expression of the different mfap4 transcripts in the different cellular population by indicating read numbers or average read numbers or any other indication of relative abundance in tables 1, 2, 3 and 4.

24) Fig 6 panel B, probably the authors forgot a horizontal line separating cells from uncut embryos versus cells from cut embryos. Such an horizontal line would help the reader understand the figure

25) Lines 342-364 Why would the authors perform the “egress” experiments through photoconversion of RBI neutrophils and photconversion of CHT neutrophils, while perform the “migration upon caudal fin transection” experiment only after “head photoconversion”.

26) Figure 7 panel A: The graphical abstract of the experimental strategy is misleading. The green arrow should stop à 44 hpf.

27) Figure 7 panel C: The graphical abstract of the experimental strategy is misleading. The green arrows for 1hpd, 2 hpd and 3 hpd should be of different lengths.

28) Line 356 "... number of CHT-derived neutrophils staying or exiting the CHT (Figure 7B)."
Do the authors really mean "exiting"?

29) Lines 513-514: "caudal fin transection was performed according to previously described (15)". Ref 15 includes two different types of caudal damage, either severe or mild. Which ones were implemented in the present work?

30) The "Materials and Methods" section reports "tracking" methods, while the results sections report "tracing" analysis. Please homogenize.

31) The "p-val" column in Table 1 is not large enough for the values to be read. For example, for "MFAP4 (1 of many).8", the value appears as "2,62E-".

32) The text refers to Tables S1, S2, S3 and S4 while we have no supplementary tables!

Point-by-point response

We appreciate the comments of the reviewers, and agree with their appreciations. We have responded point by point in the following pages. Also, we modified the manuscript to include, when appropriate, the comments raised.

Reviewer #1:

During ontogeny, multiple waves of hematopoiesis (primitive wave, EMP wave, and definitive wave) successively contribute to neutrophil generation. Those ontogenetically distinct neutrophils co-exist during certain developmental period. Yet whether those neutrophils of distinct origins manifest different cellular behaviour and function remain unclear. In this manuscript, the author made use of photoconvertible Tg(mpx:dendra2) zebrafish to label the RBI-derived (primitive wave) and the CHT-derived (EMP and definitive wave) neutrophils and monitored the behaviours and transcriptomes of mpx+ neutrophils under homeostasis and wound (fin-cut) conditions. Although the authors showed several differences of RBI- versus CHT- derived neutrophils, e.g., migration parameters and csf3b dependency, the findings (cellular motion and response to wound) lack conceptual advance and the data presented are insufficient to conclude that the RBI- and CHT-derived neutrophils are indeed differences inters of their behaviours and functions.

Major concerns:

1. Because the RBI-derived neutrophils arise earlier than CHT-derived ones, the authors should optimize their experiment settings to exclude the possibility that the cellular difference observed in their assays is not due to the difference in terms of their maturation.

Response: We thank the reviewer for this key suggestion. Regarding the time of appearance of RBI- and CHT-derived neutrophils, in our analysis we observed that the earliest Dendra2⁺ cells appeared in the RBI at 20hpf in all larvae. In the case of the CHT, some larvae showed a few Dendra2⁺ cells at 24hpf and at 30hpf they were already present in all larvae and in high numbers. Therefore, in terms of appearance of the first RBI-derived versus “late” CHT-derived cells there could be indeed an age difference. However, it has to be taken into account that RBI-derived neutrophils continue appearing after 20 hpf and thus both RBI- as well as CHT-derived populations are heterogenous in age, and eventually maturation, of the cells. Despite this age dispersion in both groups, we observed differences in the migratory behavior between RBI- and CHT-derived cells.

However, taking into account the valid comment of the author, we designed an experiment to better compare RBI- and CHT-derived cells of a similar age. To this end, we repeated the migration dynamics assay of both RBI- and CHT-derived cells (Figure 3) but 10 hours earlier (see Figure S4 A). We then first compared the migration behavior of these “younger” RBI-derived cells from this new assay with the “older” CHT-derived cells from the original assay, which should have a similar age between them. We found that they still show a difference in maximal velocity (younger RBI-derived cells: 8.00 $\mu\text{m v/s}$ older CHT-derived cells: 135.3 $\mu\text{m v/s}$; p value 0.0005) cumulative distance (younger RBI 50.0 $\mu\text{m v/s}$ older CHT-derived cells 352 $\mu\text{m v/s}$; p value 0.0001), directionality (younger RBI-derived cells: 0.15 v/s older CHT-derived cells: 0.58, p value 0.0115), and dispersion (younger RBI-derived cells: 0.521 v/s CHT-derived cells: 0.005). In the case of Euclidian distance, we did not observe differences (younger RBI-derived cells: 312.7 $\mu\text{m v/s}$ older CHT-derived cells: 290.8 $\mu\text{m v/s}$, p value 0.6406), as was the case in the original assay.

In addition, we compared the migration features of RBI- and CHT-derived cells within the assay performed 10 hours earlier. We found that both groups had the same behavior described previously in larvae 10 hours older (Figure S4). We did not observe differences in the Euclidian distance (RBI-derived cells: 312.7

$\mu\text{m v/s}$ CHT-derived cells: 316.9 μm), but we detected disparity in maximal velocity (RBI-derived cells: 8.0 $\mu\text{m/s v/s}$ CHT-derived cells: 448.8 $\mu\text{m/s}$), cumulative distance (RBI-derived cells: 50.0 $\mu\text{m v/s}$ CHT-derived cells: 442.8 μm), directionality (RBI-derived cells: 0.15 v/s CHT-derived cells: 0.54), and dispersion (RBI-derived cells: 0.521 v/s CHT-derived cells: 0.004). Indeed, for maximal velocity the same two CHT-derived cells populations were observed, a slower one comparable to the maximal velocity observer for RBI-derived cells and another much faster. Altogether, our data suggest that the differences found between RBI- and CHT-derived cells is not due to differences in age, and therefore maturation stage.

2. The conclusion “CHT-derived neutrophils have a longer lifespan compared to RBI-derived neutrophils, at least during homeostasis (Lines 184-185) is questionable. The loss of photoconverted dendra2+ (red) positive cells could be due to either cell death or excessive proliferation (the red fluorescence protein becomes diluted). Further in-depth characterization is needed to determine their lifespan.

Response: We understand the reviewer’s perspective. Thus, since we determine the period of time in which a neutrophil is present in the organism and thus can be detected, we decided to refer to this parameter in the manuscript as “period of time in which neutrophils remain in the organism”. Interestingly, a similar strategy was already used in zebrafish to quantify neutrophils lifespan (Dixon et al., 2012. doi: 10.5402/2012/915868). In this work, the authors did not distinguish between neutrophils of embryonic or definitive hematopoietic origin and used a transgenic line in which neutrophils were labeled with another photoconvertible protein, Kaede. They performed the analysis at a similar developmental stage compared to our analysis, revealing that tissue neutrophils had a half-life of approximately 5 days, similarly to our results.

3. The authors compared RBI-derived neutrophils with CHT-derived neutrophils, and they found only 5 genes upregulated in the RBI-derived cells in steady state. The *csf3b* gene is selected but with $p\text{-adjust value} = 1$ (Table 2) and the heatmap in Fig. 6A cannot clearly tell the difference of the *csf3b* expression in the RBI- and CHT-derived neutrophils. Hence, the conclusion that “RBI-derived neutrophils show higher *csf3b* expression” needs to be carefully reinvestigated.

Response: We appreciate this suggestion of the reviewer. In our DEGs analysis, the number of features (genes) differs up to 4 times between the four conditions studied (RBI-derived normal and cut, CHT-derived normal and cut). This situation impacts the identification of significant up/down regulated gene events, since in the Bonferroni correction method for multiple comparisons the $p\text{-value}$ correction directly depends on the number of features. Thus, few genes are likely to become significant up/down regulated in clusters with high number of features as is the case in our study (15,700 total features). In these situations, the use of the corrected $p\text{-value}$ for the analysis would be excessively restrictive, so many authors (Parikh et al., 2019, Supplementary Data; Travaglini, et al., 2020, Supplementary Table 4; Singh, et al., 2022, Supplementary Dataset 1, 2 and 3) considered a gene as up regulated if the $p\text{-value}$ is lower than 1, although the corrected $p\text{-value}$ is 1. To avoid this situation, we compared the number of *csf3b* normalized counts between the four conditions studied using one-way ANOVA and as post-test the Benjamini-Hochberg method. We observed a significant increase in the RBI-derived normal condition, compared to the other three comparatives (Figure S5).

4. Using *csf3b* knockdown assay, the authors showed that the egression of the RBI- but not CHT-derived neutrophils is affected. However, these data are too preliminary. Whether the egression impairment is due to the developmental defect (cell autonomous vs non-cell autonomous as well) remains unclear. The authors should examine the expression of the *csf3rb* receptors in these two ontogenically different neutrophils and how RBI- and CHT-derived neutrophils respond differently to *csf3b* signaling. These questions need to be addressed for the improvement of the manuscript quality.

Response: In a previous work, we studied the participation of Csf3b in regulating neutrophils migration during an inflammatory process in zebrafish (Galdames et al., 2014. DOI: 10.4049/jimmunol.1303220). In this context, we performed Csf3b loss-of-function assays using morpholinos and analyzed whether *csf3b* morphant embryos had developmental defects that could explain the results observed. Using different strategies, we could rule out a delay in overall development, excluded the possibility of general defects on hematopoietic cells, and confirmed the correct establishment of blood vessels. Thus, we concluded that the results observed were consequence of impaired Csf3b expression. Also, other groups have analyzed the function of Csf3b in myelopoiesis at similar developmental stages using several strategies including morpholinos, observing no general developmental disorders (Stachura et al., 2013, doi: 10.1182/blood-2012-12-475392). In the present work, we worked at similar developmental stages as in the two mentioned articles and used the same morpholino as in Galdames et al., 2014. Therefore, we think that, as in those articles, *csf3b* morphant embryos had no developmental defects in our present manuscript.

Regarding the suggestion to analyze the expression of Csf3r, as proposed, using our transcriptomic data we compared the number of *csf3r* normalized counts between the four conditions studied. We observed increased expression in the CHT-derived normal neutrophils compared to the other three groups (see figure R1). Keeping in mind that Csf3r is the receptor for Csf3a and Csf3b, and that it promotes primitive and definitive myelopoiesis, in addition to control neutrophil migration (Stachura et al., 2013), we find it difficult to correlate its expression to the results obtained in the *csf3b* morphant assay.

FIGURE REDACTED

Reviewer #2

What are the noteworthy results?

This is a very nice study, using a powerful new model to perform a detailed characterisation of phenotype and distribution of neutrophils from different developmental origins. These are characterised in detail using RNAseq. Functional profiling is followed up with single cell RNAseq analysis of CHT and RBI sorted neutrophil populations, revealing a core programme and differentially expressed genes, some of the phenotypic features depend differentially on CSF3R function, tested by knockdown.

Will the work be of significance to the field and related fields? How does it compare to the established literature? If the work is not original, please provide relevant references.

The manuscript will be of exceptional interest to zebrafish neutrophil biologists and the ultimate conclusions of interest to neutrophil biologists more generally - if some of the weaknesses can be addressed. See below. Work of this nature has not been previously performed to my knowledge. Similar experiments measuring neutrophil lifespan have been performed previously and are not cited. Dixon et al ISRN Hematology 2012:915868

Does the work support the conclusions and claims, or is additional evidence needed?

In general the experiments are well performed and the conclusions drawn support the claims made. There is one series of exceptions to this, which are discussed below.

Are there any flaws in the data analysis, interpretation and conclusions? - Do these prohibit publication or require revision?

The promoter used is a relatively short promoter and might be expected not to fully recapitulate the endogenous expression - in fact the number of macrophages identified using this approach, which do not express endogenous *mpx* would support this. This has consequences for much of the downstream analysis. The authors could investigate this by comparing *mpx* transcripts and *dendra* transcripts in the sequencing data and I would strongly suggest they show *dendra* data as a panel on Figure 4. Significant discrepancy here might undermine the generalizability of the conclusions drawn.

Response: We greatly appreciate the reviewer's enthusiasm for our manuscript. As the reviewer mentions, this is a central point in the manuscript, thus following the suggestion, we compared the number of *dendra2* normalized counts between the three identified cell groups: neutrophils, macrophages, and the "unknown" group (figure S5 A). Our analysis revealed ten times higher *dendra2* normalized counts in neutrophils compared to the macrophage and the unknown groups, (one-way ANOVA and Benjamini-Hochberg post-test, $F=1.911$, $p\text{-value}<0.001$). In neutrophils ($n=3,732$ cells), an average of 59.08 normalized counts was observed, with a median of 44.00 and a N75 of 82.00. In the case of macrophages ($n=1,830$ cells), the average was 5.51 normalized counts with a median of 2.0 and a N75 of 5.99. Finally, in the unknown group ($n=1,466$ cells) we observed an average of 2.00 with a median of 1.0 and a N75 of 1.00. Also, we determined the number of *mpx* normalized counts in the three groups and observed significantly higher normalized counts in neutrophils and nearly no counts in macrophages (Figure S5 B). Finally, we calculated the Pearson correlation coefficient between *dendra2* and *mpx* number of normalized counts, observing a high positive correlation ($r=0.69$, $p\text{-value}<0.001$). In summary, these data indicate that neutrophils (cells expressing high levels of *mpx* mRNA) express considerable higher levels of *dendra2* mRNA than macrophages and the "unknown" population (Figure S5 C). Of note, these results are in agreement with a previous report indicating that in Tg(*mpx:Dendra2*) fish, sorted *Dendra2*^{high} cells correspond to neutrophils and *Dendra2*^{low} cells to macrophage (Yoo and Huttenlocher., 2011. doi: 10.1189/jlb.1010567).

Furthermore, since the Tg(*mpx:GFP*) and Tg(*mpx:Dendra2*) lines have been shown to express high levels of the reporter proteins in neutrophils and low levels in macrophages, to further clarify that in our image analysis we focused only on neutrophils, we have added the following sentence in the results section:

“Notably, it has been shown that the Tg(mpx:GFP) and Tg(mpx:Dendra2) lines express their respective fluorescent proteins at high levels in neutrophils and at low levels in macrophages (Mathias et al., 2009 doi: 10.1016/j.dci.2009.07.003; Yoo and Huttenlocher., 2011. doi: 10.1189/jlb.1010567). Therefore, to analyze only neutrophils in our imaging experiments, we used settings that excluded cells expressing low levels of Dendra2”.

As well as a discussion of this potential weakness in this study, the authors should filter out non neutrophils before performing the analysis in figure 6. If these data are already filtered, this should be made clear in the main text and the figure legend, as currently it would appear that macrophage genes are included in the analysis. Gene ontology should also be only performed on this filtered dataset.

Response: We appreciate the reviewer raising this point to provide additional clarity. Indeed, the analysis shown in Figure 6 was performed with filtered data, taking into account only neutrophils. We modified the paragraph in which we refer to Figure 6 to make it clear that this analysis was performed only with the neutrophil population as follow: *“To this end, we performed differential gene expression analyses between all the four groups but using only the data from the cluster corresponding to neutrophils...”*

Minor issues

Line 246 - CHT neutrophils are a mixture of CHT and RBI neutrophils, so maybe this explains why there are 2 different populations? Also relevant to line 447. CHT cells more heterogenous in behaviour. Varying degrees of development as well as containing 2 populations - RBI cells migrated into CHT and native CHT cells.

Response: We thank the reviewer for raising this point which is a possibility that is difficult to rule out. However, based on our observations, we believe the RBI-derived neutrophils that potentially localized in the CHT would be too few to explain the group of cells in the CHT-derived neutrophils that display a similar behavior to RBI-derived cells in the functional assay (Figure 3 D, E). The percentage of RBI-derived neutrophils that colonized the CHT during a period of 72h was 6.25% (2 cells from a total of 32). A similar percentage of CHT-derived neutrophils (8,33%, 2 cells from a total of 24) colonized the head in the same 72h period, but we still observed a homogeneous response in the RBI-derived neutrophil functional assays. On the other hand, based on the literature (Murayama., et al., 2006; Bertrand et al., 2007) at the developmental stages we performed our analysis, CHT-derived neutrophils are composed of neutrophils originated from erythromyeloid progenitors (EMP) and hematopoietic stem cells (HSC) present in the CHT. Taking this into account, it is quite possible that the difference in behavior observed could be explained by the presence of these two ontogenically distinct populations. We included these comments in the discussion section.

Line 440. RBI neutrophils only migrate through tissue, while CHT migrate through vessels. Obvious anatomical explanations for this which are not discussed.

Response: We thank to the reviewer for the comment. Yes, since CHT-derived neutrophils are mainly located in the mesenchyme around the caudal vein plexus, they have more chances of entering the bloodstream compared to RBI-derived neutrophils that are preferably located in the subepidermal mesenchyme of the head. We included this comment in the discussion section as follow: *“This situation could be a consequence of differences in the anatomy of the tissues in which they are located. In the CHT, neutrophils are mainly found in the mesenchyme around the caudal vein plexus, i.e. in close proximity to the vasculature. In contrast, RBI-derived neutrophils are preferably located in the subepidermal mesenchyme of the head, more distant from blood vessel”*.

Line 457 “a couple of genes”. Suggest: “several”

Response: We incorporated the suggestion and changed the sentence, thanks.

Line 457 - the new granulocyte genes mentioned would be stronger if cross-referenced to one of the many human RNAseq datasets available.

Response: Thank you very much for the comment. We have searched for *il34*, *il4*, *ponzr6* and *npsn* expression in public repositories of human single cell RNA sequencing such as "Human Cell Data Portal", "Single Cell Expression Atlas from the EMBL's European Bioinformatics Institute" and DISCO. Unfortunately, we were unable to draw robust conclusions. One of the main problems was that there is no uniform way of labeling the different cell types. For example, we found groups labeled as "neutrophils," "immature/mature neutrophils," and "granulocytes," which makes comparison with our data sets difficult. In addition, we found low expression levels of genes that we identified as neutrophil markers in the datasets, including the few in which neutrophils and macrophages were annotated as such. Finally, most datasets from humans are from immune cells isolated from circulating blood while our analysis is from cells located in tissues.

The colours in Figure 5 are difficult to discern and could be made clearer.

Response: We thank the reviewer for pointing out this issue. We have changed the code color for CHT-derived neutrophils normal and cut (green and light green respectively), as well as for RBI-derived neutrophils normal and cut (red and pink respectively). We hope this color-code is more clear to the reader.

Line 295 - give rise to...is perhaps misleading. The RBI cells are exclusively neutrophils, while the CHT cells either are or give rise to.

Response: Thanks for the comment from the reviewer, we have rephrased the sentence.

Line 297 more pronouncedly is grammatically incorrect.

Response: Thanks for noting this grammar issue, we have replaced it by "more pronounced".

Figure 7 - a Mann-Whitney U test would need to be corrected for multiple comparisons.

Response: We thank the reviewer for this observation, we apologize for this mistake. We have used Kruskal-Wallis Test for analyzing the data in this figure. We corrected the information in the figure legend.

Reviewer #3

The manuscript « Ontogenetically distinct neutrophils differ in behavior and transcriptional profiles in zebrafish » by Juan P. García-López, Alexandre Grimaldi, Zelin Chen, Karina Bravo-Tello¹, Erica Bresciani, Alvaro Banderas, Shawn M. Burgess, Pedro P. Hernandez and Carmen G. Feijóo addresses the interesting problem of the specificities of the neutrophils from the primitive versus the definitive wave. This question is clearly of importance for the field, and more broadly in the field of early innate immunity. They chose to address this important question using the zebrafish model, and the results presented in this manuscript suggest that these two neutrophil populations may have different flavors.

I have two categories of concerns:

Main concerns to be addressed:

1) Macrophages versus neutrophils: The “transcriptomic” part of the manuscript reveals that it is probable that the so called CHT-derived neutrophils studied in the first three sections of the results are in fact both macrophages and neutrophils (roughly 50%/50%) severely compromising the conclusions. For the study of their CHT-derived neutrophils as reported in these first three sections of the results, the authors shall therefore implement new strategies to discriminate macrophages versus neutrophils.

Response: We highly appreciate the comments of the reviewer and useful feedback of our manuscript. It has been indeed reported that in Tg(mpx:Dendra2) fish, neutrophils express high levels of Dendra2 thus being strongly fluorescent, and that macrophages express low Dendra2 levels, consequently showing slight fluorescence (Yoo and Huttenlocher., 2011. doi: 10.1189/jlb.1010567). Similarly, the Tg(mpx:GFP) has been shown to express high levels of the reporter proteins in neutrophils and low levels in macrophages (Mathias et al., 2009 doi: 10.1016/j.dci.2009.07.003). Taking into account these observations, we have indeed used imaging settings that allowed us to analyze only neutrophils and not macrophages in our imaging experiments (Figure R2 below). We apologize for not having provided details in this regard in the first version of the manuscript. To further clarify that in our image analysis we focused only on neutrophils, we have added the following sentence in the results section: “Notably, it has been shown that the Tg(mpx:GFP) and Tg(mpx:Dendra2) lines express their respective fluorescent proteins at high levels in neutrophils and at low levels in macrophages (Mathias et al., 2009 doi: 10.1016/j.dci.2009.07.003; Yoo and Huttenlocher., 2011. doi: 10.1189/jlb.1010567). Therefore, to analyze only neutrophils in our imaging experiments, we used settings that excluded cells expressing low levels of Dendra2”.

FIGURE REDACTED

2) Age of the different populations of neutrophils:

The first neutrophils of the primitive wave are present in the RBI at 20 hpf and may be detected as soon as 16 hpf while those of the second wave are not detected before 30 hpf. When the authors do photoconvert neutrophils from both waves at 34 + 44 hpf, and analyze them at the same time point, the respective RBI and CHT derived neutrophils do not have the same “age”; the RBI derived ones are “older” ones. The authors should consider their age differences and set up experimental frames to integrate this difference to measure their respective migration capacity, lifespan, H2O2 dependent migration, migration dynamics and parameters, and transcriptomic profiles.

Response: We thank the reviewer for this key suggestion also identified by reviewer 1. Regarding the time of appearance of RBI- and CHT-derived neutrophils, in our analysis we observed that the earliest Dendra2⁺ cells appeared in the RBI at 20hpf in all larvae. In the case of the CHT, some larvae showed a few Dendra2⁺ cells at 24hpf and at 30hpf they were already present in all larvae and in high numbers. Therefore, in terms of appearance of the first RBI-derived versus “late” CHT-derived cells there could be indeed an age difference. However, it has to be taken into account that RBI-derived neutrophils continue appearing after 20 hpf and thus both RBI- as well as CHT-derived populations are heterogenous in age, and eventually maturation, of the cells. Despite this age dispersion in both groups, we observed differences in the migratory behavior between RBI- and CHT-derived cells.

However, taking into account the valid comment of the author, we designed an experiment to better compare RBI- and CHT-derived cells of a similar age. To this end, we repeated the migration dynamics assay of both RBI- and CHT-derived cells (Figure 3) but 10 hours earlier (see Figure S4 A). We then first compared the migration behavior of these “younger” RBI-derived cells from this new assay with the “older” CHT-derived cells from the original assay, which should have a similar age between them. We found that they still show a difference in maximal velocity (younger RBI-derived cells: 8.00 $\mu\text{m v/s}$ older CHT-derived cells: 135.3 μm ; p value 0.0005) accumulative distance (younger RBI 50.0 $\mu\text{m v/s}$ older CHT-derived cells 352 μm ; p value 0.0001), directionality (younger RBI-derived cells: 0.15 v/s older CHT-derived

cells: 0.58, p value 0.0115), and dispersion (younger RBI-derived cells: 0.521 v/s CHT-derived cells: 0.005). In the case of Euclidian distance, we did not observe differences (younger RBI-derived cells: 312.7 μm v/s older CHT-derived cells: 290.8 μm , p value 0.6406), as was the case in the original assay. In addition, we compared the migration features of RBI- and CHT-derived cells within the assay performed 10 hours earlier. We found that both groups had the same behavior described previously in larvae 10 hours older (Figure S4). We did not observe differences in the Euclidian distance (RBI-derived cells: 312.7 μm v/s CHT-derived cells: 316.9 μm), but we detected disparity in maximal velocity (RBI-derived cells: 8.0 $\mu\text{m/s}$ v/s CHT-derived cells: 448.8 $\mu\text{m/s}$), accumulative distance (RBI-derived cells: 50.0 μm v/s CHT-derived cells: 442.8 μm), directionality (RBI-derived cells: 0.15 v/s CHT-derived cells: 0.54), and dispersion (RBI-derived cells: 0.521 v/s CHT-derived cells: 0.004). Indeed, for maximal velocity the same two CHT-derived cells populations were observed, a slower one comparable to the maximal velocity observer for RBI-derived cells and another much faster. Altogether, our data suggest that the differences found between RBI- and CHT-derived cells is not due to differences in age, and therefore maturation stage.

3) Origin versus homing:

The RBI is the place where primitive neutrophils do appear, and this is fundamentally different from the CHT that is a homing -... tissue where HSC from the AGM do differentiate. It is therefore not relevant to compare their capacity to "retain neutrophils".

Response: We thank the reviewer for commenting on this very interesting point. We think that a point to be taken into account is that at this developmental stage the CHT has actually both functions, origin and homing. Erythromyeloid progenitors (EMP) arise and differentiate there, and hematopoietic stem cells (HSC) migrate to the CHT to differentiate (Bertrand et al., 2007; Murayama., et al., 2006). Thus, at the developmental stage we performed our experiments EMP-derived and HSC-derived neutrophils were present in the CHT. On the other hand, we performed experiments that help answering the reviewer comment, see Figure S1 A-C for RBI and Figure 1D for CHT. The initial propose of these assays was to analyze the migration capacity of RBI-derived and CHT-derived neutrophils in the absence of an insult. Here, those cells that did not migrate, can be considered as "retained" in the tissue where they were located. We observed that 24h after photoconverted the RBI, 97,07% of the RBI-derived neutrophils where outside this region. On the other hand, 72h after photoconverted the CHT, 50% of the CHT-derived cells were still in the CHT. Thus, these results make the point that RBI is the origin tissue for RBI-derived neutrophils but that they will quickly leave it. In contrast, several of the CHT-derived neutrophils stay there for at least 72h.

4) Single cell transcriptomics:

While for the first three sections of the results, the authors have work with both RBI- and CHT-photoconverted cells, in the "transcriptomic" section, for obvious financial and practical reasons, the authors have decided that photoconverted cells are from RBI origin while non photoconverted ones are from CHT origin. If the corresponding results are true, them half of the CHT-derived cells analyzed in the previous parts of the article are in fact macrophages while only a small fraction of the RBI-derived cells are macrophages. This could interfere severely with the conclusions draw from data presented in figures 1 to 3. It is therefore necessary to perform this transcriptomic analysis with CHT-photoconverted neutrophils not only to be sure that the population labelled as "CHT-derived" is of true CHT origin, but also to know the proportion of macrophages in the CHT-derived populations analyzed in the previous sections of the manuscript (figures 1 to 3)

Response: We thank the reviewer for this observation. As mentioned above, in Tg(mpx:Dendra2) fish neutrophils are Dendra2^{high} and macrophages Dendra2^{low}. Thus, if CHT-derived Dendra2⁺ cells are photoconverted or not, the group will be composed by neutrophils (high level of dendra2 mRNA) and macrophage (low level of dendra2 mRNA). To corroborate that this situation was true in our

transcriptomic analysis, we compared the number of *dendra2* normalized counts between the three identified cells groups; neutrophils, macrophages, and the unknown group (Figure S5). The results showed ten times higher *dendra2* normalized counts in neutrophils compared to macrophage and the unknown group, (one-way ANOVA and Benjamini-Hochberg post-test, $F=1.911$, $p\text{-value}<0.001$). In neutrophils ($n=3,732$ cells), an average of 59.08 normalized counts was observed, with a median of 44.00 and a N75 of 82.00. In the case of macrophages ($n=1,830$ cells), the average was of 5.51 normalized counts with a median of 2.0 and a N75 of 5.99. Finally, in the unknown group ($n=1,466$ cells) we observed an average of 2.00 with a median of 1.0 and a N75 of 1.00. Also, we determined the number of *mpx* normalized counts in the three groups and observed significantly more normalized counts in neutrophils and nearly no counts in macrophages. Finally, we calculated the Pearson correlation coefficient between *dendra2* and *mpx* number of normalized counts, observing a high positive correlation ($r=0.69$, $p\text{-value}<0.001$). We hope that these arguments, together with our response to the reviewer's point 1 regarding how we excluded macrophages and focused on neutrophils in our imaging analysis are satisfactory.

Minor concerns to be addressed:

5) Figure to highlight what RBI is! It would be nice to have a first figure panel for this.

Response: We thank the reviewer for the comment. We included a panel showing RBI location in Figure S1.

6) Lines 130-131: To my opinion, reference 23 is not relevant for the assertion "On the other hand, CHT-derived neutrophils mainly remain in the caudal hematopoietic tissue (23)". The work presented in the manuscript shows that this assertion is correct for the first 48 hours of development.

Response: We thank the reviewer for this comment and we apologize for misplacing the citation. We replaced it by the reference Bertrand et al., 2007 (doi: 10.1242/dev.012385) in which it was demonstrated that cells at the CHT at 40hpf remained there 4 days later.

7) Line 147: "...new green neutrophils appeared in this area..." it is not clear what "this area" is. Is it the head or the CHT? Please clarify in the text.

Response: Thanks to the reviewer for noting this ambiguity. We have replaced "in this area" by "head"

8) Lines 146-150: This part is dedicated to setting up the photoconversion frame so that a maximum of RBI derived neutrophils would be labelled. If I understood well, the authors show that green neutrophils that appear in the head between 34 and 44 hpf do come from the RBI (primitive wave). Therefore, green neutrophils present in the head at 44hpf in panel S1E, are not "CHT-derived neutrophils". Panel S1E shall therefore be changed and "CHT-derived neutrophils" replaced by "green neutrophils".

Response: Many thanks for the comment. To avoid confusing the reader, we modified the panel S1E (now S2 E) and only left the data showing the appearance of new green *Dendra2*⁺ cells in the head at 44hpf, which was the purpose of this data set (panel S1 D-F, now S2 D-F).

9) Figure Sup1:

The size of the fonts is too small, it is difficult to read, and when enlarged, they pixelize.

The ordinate of panel B is absolutely not adapted for two reasons: 1) For 20 hpf, as the count of neutrophils is 0, it is not possible to use a % scale. 2) panels E and H are with absolute numbers, it is necessary to homogenize so that the reader could compare. Please use absolute numbers for panels B, E & H.

Response: We thank the reviewer for this comment to ameliorate the display of the figure. We changed the figure and used absolute numbers for the graph in panel B.

The color code for panel E does not allow to recognize RBI versus CHT derived cells. The dots for RBI derived neutrophils appear black, it would sound obvious that they would be red.

Panel H: The ordinate is labelled “number of neutrophils” while it should be labelled “number of red neutrophils”... and it would be much easier to understand with red dots.

Response: Many thanks for the comment of the reviewer. We labelled the ordinate as “number of red neutrophils”. The dots were red but we have lightened the color to make it more evident.

10) Lines 160-162: For this part of their work, the authors consider that green macrophages are CHT-derived while red ones are RBI derived. This is not acceptable because the authors have explained in the previous paragraph that green cells in the head at 44hpf were in fact of RBI origin (lines 147-150)

Figure 1B shall therefore be changed.

Response: Many thanks for the comment and we apologize for the confusing labelling. In this figure, either the head or CHT was photoconverted and the resulting red cells were followed. We changed the colors of the bars in the graph in panel B to avoid a confusion with red and green neutrophils.

11) Line 161 “photoconverted the head or the at 34 hpf and analyzed at 44 hpf”, it seems that the authors forgot at least a word after “or the”.

Response: We apologize for the mistake. We corrected the sentence and included the missing word “CHT”.

12) Panel 1C is supposed to be a graphical abstract of the time frame of the experiment, but the scale is hpf while data 1D to 1I are at 72 hours post second photoconversion (hpp). Please indicate 72 hpp in panel 1C.

Response: Many thanks for the comment. We have incorporated an arrow to indicate the 72hpp.

13) Panel 1E, “neutrophils” should be replaced by “CHT-derived neutrophils”

Response: Many thanks for the comment, we have now labelled the ordinate as “CHT-derived neutrophils”

14) Panel 1G, “neutrophils” should be replaced by “RBI-derived neutrophils”

Response: Many thanks for the comment, we labelled the ordinate as “RBI-derived neutrophils”

15) Figure 2 panel A (same in S2A): The graphical abstract of the experimental strategy is misleading. The green arrows for 1hpd, 2 hpd and 3 hpd should be of different lengths.

Response: Thanks to the reviewer for the comment, we made green arrows of different lengths.

16) Line 214 “... diphenyliodonium (DPI), an inhibitor of NADPH oxidase that inhibits hydrogen peroxide signaling”; DPI does not inhibit signaling, it inhibits production of H₂O₂.

Response: We apologize for the mistake. We replaced “signaling” by “production”.

17) Line 215 “... and therefore the recruitment of neutrophils to wound”; No! the sentence should be “... and therefore the H₂O₂ dependent recruitment of neutrophils to wound”

Response: We corrected the sentence according to the suggestion of the reviewer.

18) Lines 263-339: The exact time frame of the experiment is lacking. The time of transection is lacking, and the number of hours between damage and homogenization of embryos for cell sorting is also lacking.

Response: We apologize for the missing information. We added the missing data in the Material & Methods section, tail transection was performed at 54hpf and homogenization was carry out at 57hpf.

19) Lines 291-292 “Interestingly, while neutrophils contained a similar percentage of cells from all four conditions, macrophages mainly comprised CHT-derived cells (Figure 5B)”. This sentence is not understandable! It should be rephrased.

Response: We thank the reviewer for the comment. We rephrased the paragraph as follows: “Interestingly, while neutrophils were present in similar quantities in the four experimental conditions, macrophages were mainly present in CHT-derived cells. This implies that RBI-derived cells contributed mainly to the neutrophil population, while the CHT-derived cells contributed in a similar proportion to the neutrophil and macrophage populations”.

20) Line 295 “..., mp x :Dendra2+ RBI-derived cells give rise almost exclusively to neutrophils ...” No! The statement should be “..., at 34 and 44 hpf mp x :Dendra2+ cells in the RBI are give rise almost exclusively neutrophils ...”

Response: We thank the reviewer for detecting this mistake. We have rephrased the sentence as follows: “...mp x :Dendra2⁺ cells originated in the RBI that arise in the head between 34 and 44 hpf are almost exclusively neutrophils....”

21) Line 296 the statement “...mp x :Dendra2+ CHT-derived cells give rise to neutrophils and macrophages in similar proportions.” is wrong! This is due to the fact that the authors consider all green cells as CHT derived.

Response: We understand the reviewer's point of view. Based on our data and in our opinion, the colonization of the CHT by RBI-derived cells is too low to explain the results observed in the transcriptomic analysis (50% neutrophils and 50% macrophages). We observed that between 44hpf and 116hpf, a period of 72h, 2 cells from a total of 32 (6,25%) originally present in the head colonize the CHT. If we consider that the transcriptomic analysis was made 13h post second photoconversion, it could be that RBI-derived cells present in the CHT were even less than the 6,25% observed after the 72h. Of note, during these 13h we observed a maximum of 1 new green neutrophil appearing in the head or RBI in some of the analyzed larvae. Taking the aforementioned argument into account, we think that the most probable scenario is that mp x :Dendra2⁺ CHT-derived cells contain macrophages in addition to the expected neutrophils.

22) Line 297 “Upon inflammation, macrophages increase more pronouncedly than neutrophils...” What exactly the authors mean by “increase” is it cell size or volume or is it numbers? ... does this mean that macrophages have a longer lifespan or that they replicate while neutrophils don't and even die?

Response: We thank the reviewer for the comment. We meant that the number of macrophages present in the groups of Dendra2+ cells analyzed increase more than the number of neutrophils. The explanation for this fact is most likely due to several reasons, including those mentioned by the reviewer. Despite this is a very interesting point, it is beyond the scope of our current work. We rephrased the sentence to state the idea more clearly as follows: “Upon inflammation, the number of macrophages increase more pronounced than the number of neutrophils, with RBI-derived macrophages increasing the most”.

23) Line 280 “The second largest cluster expressed significantly higher levels of 186 genes, including mfap4...” while in tables 2 and 3 the authors show that the different populations of neutrophils express different levels of mfap4. As the authors highlight, there are many different mfap4 transcripts, it is therefore necessary that the authors refer to reference zebrafish mfap4 transcripts so that the reader would know exactly which transcript is being considered. Furthermore, as the mfap4 expression is considered as an excellent marker for macrophages, it is surprising that abundance of mfap4 transcripts

could vary between different neutrophil populations. It would be very helpful for the reader to have an idea of the relative expression of the different *mfap4* transcripts in the different cellular population by indicating read numbers or average read numbers or any other indication of relative abundance in tables 1, 2, 3 and 4.

Response: Thanks for the comment to the reviewer. We included in the sentence the name of six *mfap4* transcript upregulated in macrophage, *mfap4.1*, *mfap4.2*, *mfap4.4*, *mfap4.7*, *mfap4.8*, *mfap4.10*.

24) Fig 6 panel B, probably the authors forgot a horizontal line separating cells from uncut embryos versus cells from cut embryos. Such an horizontal line would help the reader understand the figure

Response: Many thanks for the comment. In Fig 6 panel B, as in panel C, genes that changed their expression before and after tail transection are showed. We have incorporated the suggested changes in the both panels.

25) Lines 342-364 Why would the authors perform the “egress” experiments through photoconversion of RBI neutrophils and photconversion of CHT neutrophils, while perform the “migration upon caudal fin transection” experiment only after “head photoconversion”.

Response: We thank the reviewer for the question. The purpose of the egress experiment was to quantify the number of Dendra2⁺ cells that left the RBI during a defined period of time, thus the only suitable experimental setting was to perform the photoconversion directly at the RBI. On the other hand, we realized that to analyze neutrophil recruitment during inflammation, the number of photoconverted cells in the RBI was too low, as we determined that Dendra2⁺ cells in the RBI rapidly leave this territory and accumulate in the head. Therefore, we decided to perform photoconversion in the head to have a higher number of labeled neutrophils for further analysis.

26) Figure 7 panel A: The graphical abstract of the experimental strategy is misleading. The green arrow should stop à 44 hpf.

Response: We apologize for the mistake, we have corrected it.

27) Figure 7 panel C: The graphical abstract of the experimental strategy is misleading. The green arrows for 1hpd, 2 hpd and 3 hpd should be of different lengths.

Response: Thanks for the comment, we made green arrows of different lengths.

28) Line 356 “... number of CHT-derived neutrophils staying or exiting the CHT (Figure 7B).” Do the authors really mean “exiting”?

Response: Thanks for the comment, we apologize for the mistake. It has been corrected.

29) Lines 513-514: “caudal fin transection was performed according to previously described (15)”. Ref 15 includes two different types of caudal damage, either severe or mild. Which ones where implemented in the present work?

Response: Thanks for the comment regarding detailing the transection protocol used. We rephrased the sentences as follows: “...then a severe caudal fin transection model was performed according to previously described (15)”

30) The “Materials and Methods” section reports “tracking” methods, while the results sections report “tracing” analysis. Please homogenize.

Response: We thank the reviewer for the comment. We replaced the word “tracking” by “tracing” in the whole manuscript.

31) The “p-val” column in Table 1 is not large enough for the values to be read. For example, for “MFAP4 (1 of many).8”, the value appears as “2,62E-”.

Response: Thanks for the comment, we apologize for the mistake. We have enlarged the width of each column to make it easier to read.

32) The text refers to Tables S1, S2, S3 and S4 while we have no supplementary tables!

Response: Thanks for the comment, we apologize for the mistake. We changed the name of the tables according to the text.

REVIEWERS' COMMENTS

Reviewer #1 (Remarks to the Author):

The revised manuscript has addressed majority of my concerns. However, the mechanism underlying the difference of RBI- vs CHT-derived neutrophils egression in response to csf3b-csf3r signaling is still lacking.

Reviewer #2 (Remarks to the Author):

The authors have provided a detailed rebuttal to comments from myself and the other two reviewers. I am satisfied that they have addressed these points to an appropriate level.

Reviewer #3 (Remarks to the Author):

I consider that the authors have correctly addressed the issues raised by the referees.
Congratulations for this nice piece of work.
Georges Lutfalla

Second Point-by-point response to reviewers

We acknowledge the feedback provided by the three reviewers. Regarding the last comment of reviewer 1, please find our response below.

Reviewer #1:

The revised manuscript has addressed majority of my concerns. However, the mechanism underlying the difference of RBI- vs CHT-derived neutrophils egression in response to csf3b-csf3r signaling is still lacking.

R: Thanks for the comment. In mammals, several studies have demonstrated the importance of specific residues in particular domains of CSF3R undergoing posttranslational modifications to activate or inhibit signaling in this pathway (Aarts et al., 2004, doi: 10.1182/blood-2003-07-2250; Wolfler et al., 2009, doi: 10.1111/j.1600-0854.2009.00928.x; Irandoust et al., 2007, doi: 10.1038/sj.emboj.7601640). Therefore, we believe that similar regulatory mechanisms could exist in fish. Indeed, there is evidence that at least the Jak2 and Pi3k signaling pathways are present (Meier et al., 2022, doi: 10.31083/j.fbl2704110; Meenhuis et al., 2009, doi: 10.1042/BJ20081153; Nakamae-Akahori et al., 2006, doi: 10.1111/j.1365-2567.2006.02448.x). In this scenario, it could be that CHT-derived neutrophils, even in the presence of Csf3b, do not respond by migrating due to inhibition of Csf3r. Consequently, the signaling pathway would not be activated.

We have included a comment on this in the discussion, highlighted in red.